# On the Theoretical Limitations of Embedding-based Link Prediction

**Samy Badreddine** [1 2 3]  **Emile van Krieken** [* 4]  **Luciano Serafini** [* 2 3]

## Abstract

Neural networks often map low-dimensional embeddings to high-dimensional output spaces. Usually, the output layer is linear, which can create a *rank bottleneck* that limits the functions a model can represent. Such bottlenecks are ubiquitous in link prediction models, such as knowledge graph embeddings (KGEs), as the output space of entities can be orders of magnitude larger than the embedding dimension. We investigate how rank bottlenecks limit model expressivity for fitting the training data. While previous work focused on sufficient bounds on the embedding dimension required for specific KGEs, we show necessary bounds for *all* KGEs with a linear output layer, which grow with graph size and connectivity. We also consider a non-linear output layer using mixtures to break the bottleneck without significant parameter overhead. Empirically, we show that models using this non-linear layer improve in ranking performance and probabilistic fit for large and dense datasets, as predicted by our theory. Our work reveals how linear output layers limit KGEs and motivates non-linear alternatives for scaling to large and dense graphs.

## 1. Introduction

Mapping entities and relations into low-dimensional vectors is central to modern link prediction. Mostly explored in knowledge graph (KG) research, link prediction can reveal drug-disease associations (Himmelstein et al., 2017; Bonner et al., 2022; Daza et al., 2023) or social network and e-commerce recommendations (Hu et al., 2020). For scalability, most knowledge graph embedding (KGE) models use a linear output layer to score a subject–relation query against objects. This holds for all bilinear and most neural approaches, even when they use powerful encoders like graph neural networks or language models (Ali et al., 2022). Such linear output layers introduce low-rank constraints, often referred to as *rank bottlenecks* (Yang et al., 2018; Grivas et al., 2024). Concretely, when the number of target entities (typically, $10^5$ to $10^9$ for industry-scale KGs (Sullivan, 2020)) exceeds the embedding dimension ($10^2$ to $10^4$), the model expressivity is limited by the rank of its output layer.

Previous works have shown ranking expressivity bounds for specific bilinear KGEs for link prediction (Trouillon et al., 2017; Wang et al., 2018; Balazevic et al., 2019), but such bounds are missing for more general KGEs and for other prediction tasks. We use insights from linear algebra and existing work on rank bottlenecks to derive dimensional requirements. We do so for three tasks aimed at either (i) expressing all possible patterns in graphs of a certain size (*universality*) or (ii) expressing specific patterns in a given graph (*realisability*). The required dimensions grow with graph size and connectivity.

Because these requirements are impractical for large and dense KGs, we propose breaking the bottleneck by replacing the linear output layer with a non-linear one. We introduce KGE-MoS, a drop-in non-linear output layer using a mixture of softmaxes (MoS) (Yang et al., 2018). We evaluate KGE-MoS on five KGE models across KGs of increasing size. In larger and denser regimes, KGE-MoS improves both ranking accuracy and probabilistic fit at a much lower parameter cost than simply increasing the embedding dimension.

**Contributions.** We (i) derive necessary bounds for universality and realisability on three prediction tasks in bilinear and neural KGEs (Section 3). We (ii) give sufficient dimensions to fit some tasks in relation to graph connectivity (Section 4). We (iii) introduce KGE-MoS, a non-linear output layer that breaks bottlenecks and improves performance in large benchmarks at a low parameter cost (Section 5).

Our results suggest link prediction should move beyond encoders by also exploring more expressive output layers.

---

*Equal supervision  [1]Sony AI, Japan  [2]Fondazione Bruno Kessler, Italy  [3]University of Trento, Italy  [4]Vrije Universiteit Amsterdam, Netherlands. Correspondence to: Samy Badreddine <samy.badreddine@sony.com>, Emile van Krieken <e.van.krieken@vu.nl>, Luciano Serafini <luciano.serafini@fbk.eu>.

*Proceedings of the $43^{rd}$ International Conference on Machine Learning*, Seoul, South Korea. PMLR 306, 2026. Copyright 2026 by the author(s).

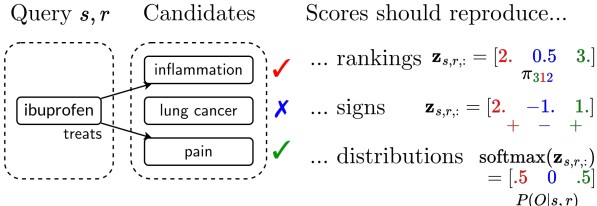

*Figure 1.* KGEs are used in various prediction tasks, each with their own expressivity needs.

## 2. Background

### 2.1. Rank Bottlenecks in Output Layers

Complex inference tasks often involve mapping low-dimensional data embeddings to high-dimensional output spaces. When the mapping is linear, this creates low-rank constraints that limit the model's expressivity. Concretely, consider a hidden vector $\mathbf{h} \in \mathbb{R}^d$ that is projected by a linear layer $\mathbf{W} \in \mathbb{R}^{m \times d}$ to an output vector $\mathbf{z} = \mathbf{h}\mathbf{W}^\top \in \mathbb{R}^m$. Regardless of the expressivity of the upstream encoder, the output $\mathbf{z}$ is confined to the column space of $\mathbf{W}$, which has dimension at most $d$. In other words, the model can only produce outputs lying in a $d$-dimensional linear subspace of the $m$-dimensional output space. When $d \ll m$, this restriction is known as a *rank bottleneck*.

Rank bottlenecks are typically benign in intermediate layers because repeated non-linear transformations can "reshape" the linear subspace into meaningful manifolds, like in autoencoders (Hinton & Zemel, 1993). However, at the final layer, where predictions are consumed directly by, e.g., a softmax or sigmoid layer, this constraint imposes rigid limitations. This phenomenon has been well-studied in language modelling (Yang et al., 2018; Grivas et al., 2022). However, rank bottlenecks are not yet well-explored in link prediction, even though the output space of entities ranges from tens of thousands to millions and is often far larger than common language model vocabularies.

### 2.2. Knowledge Graph Embeddings

A knowledge graph (KG) is a common data structure for representing relational information. Let $\mathcal{E}$ be a set of entities and $\mathcal{R}$ a set of relations. A KG $\mathcal{G} = \{(s, r, o) \in \mathcal{E} \times \mathcal{R} \times \mathcal{E}\}$ is a set of triples where $s$ is a subject, $r$ is a relation, and $o$ is an object. In link prediction, only some triples $\mathcal{G}_{\text{obs}} \subset \mathcal{G}$ are observed, and the task is to predict missing triples $\mathcal{G} \setminus \mathcal{G}_{\text{obs}}$ by predicting suitable objects for $(s, r, ?)$ queries. This is typically cast as a ranking problem where candidate objects are scored and ranked; correct answers are objects $o$ such that $(s, r, o) \in \mathcal{G}$ (Nickel et al., 2015). While our experiments are on KGs, our mathematical theory is applicable to all embedding-based link prediction.

A knowledge graph embedding model (KGE )is a scoring function $\phi : \mathcal{E} \times \mathcal{R} \times \mathcal{E} \to \mathbb{R}$ that assigns a score $z_{s,r,o}$ to each triple by embedding entities and relations.[1] Here, higher scores correspond to more likely triples. Let $\mathcal{Z} \in \mathbb{R}^{|\mathcal{E}| \times |\mathcal{R}| \times |\mathcal{E}|}$ denote the score tensor with entries $z_{s,r,o}$. A KGE answers object prediction queries $(s, r, ?)$ by ordering entities by decreasing values of $\mathbf{z}_{s,r,:} \in \mathbb{R}^{|\mathcal{E}|}$.

#### 2.2.1. LINEAR OBJECT PREDICTION

In this work, we consider KGEs using a dot product scoring function between two $d$-dimensional embeddings:

$$\phi(s, r, o) = \mathbf{h}_{s,r}^\top \mathbf{e}_o. \tag{1}$$

Here, $\mathbf{e}_o \in \mathbb{R}^d$ is an embedding of the object $o$, and $\mathbf{h}_{s,r} \in \mathbb{R}^d$ is an embedding of the subject and relation, which are typically results of some encoding function.[2] This class of KGEs linearly scores object embeddings, and hence we call it *linear-decoder KGEs (LD-KGEs)*. Let $\mathbf{E} \in \mathbb{R}^{|\mathcal{E}| \times d}$ be the embeddings of all entities stacked in a matrix. Computing the scores of a fixed subject-relation pair $(s, r)$ against all object entities is the efficient vector-matrix multiplication

$$\mathbf{z}_{s,r,:} = \mathbf{h}_{s,r} \mathbf{E}^\top. \tag{2}$$

LD-KGEs include, as their name suggests, all *bilinear KGEs* such as RESCAL (Nickel et al., 2011), DISTMULT (Yang et al., 2014), COMPLEX (Trouillon et al., 2017), CP (Lacroix et al., 2018), SIMPLE (Kazemi & Poole, 2018), or TUCKER (Balazevic et al., 2019). These models, despite being among the earlier KGE methods, remain particularly popular in large-scale applications for their scalability.

Most *Neural KGEs* are also of the form in Equation (1) and hence in LD-KGEs. They use a powerful neural encoder to embed subject and relation queries into a hidden state $\mathbf{h}_{s,r} = \text{NeuralNet}(s, r)$. Then they project the hidden state to logits in the object space using a simple linear layer, which recovers $\mathbf{h}_{s,r}\mathbf{E}^\top$. Neural encoders can be CNNs like CONVE (Dettmers et al., 2018; Balažević et al., 2019), graph convolutional networks like COMPGCN (Schlichtkrull et al., 2018; Vashishth et al., 2020), transformers (Galkin et al., 2020) or language models (Wang et al., 2022a; Choi et al., 2021; Wang et al., 2022b; Liu et al., 2022). More recently, many neural methods use cosine similarity (Lin et al., 2024; Li et al., 2024; Shan et al., 2024; Yang et al., 2024). They also fall under our theory.

We discuss the relevant models in detail in Appendix D.1. In Section 7, we discuss link prediction methods that do not use the dot product and are outside the scope of this work.

---

[1] We use $a$, $\mathbf{a}$, $\mathbf{A}$, $\mathcal{A}$ to denote scalars, vectors, matrices, and tensors respectively. See Appendix A for indexing notation.

[2] We also consider KGEs using the cosine similarity as the scoring function, as one can redefine $\mathbf{h}'_{s,r} = \mathbf{h}_{s,r}/\|\mathbf{h}_{s,r}\|$ and $\mathbf{e}'_o = \mathbf{e}_o/\|\mathbf{e}_o\|$ to retrieve the dot product.

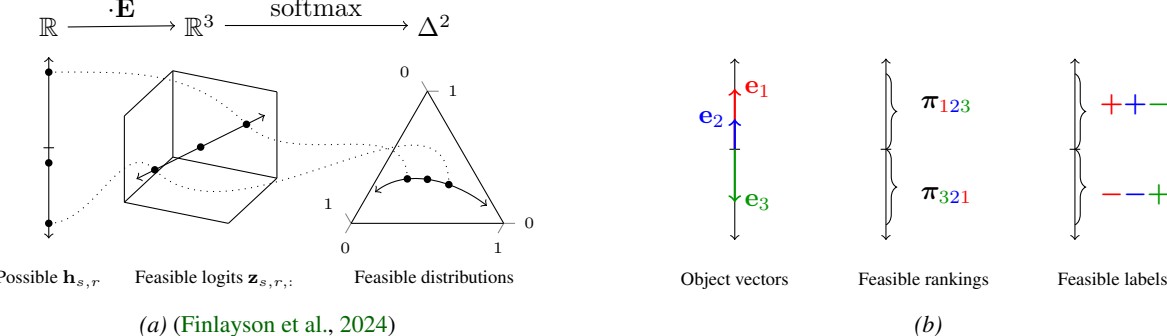

*(a)* (Finlayson et al., 2024)            *(b)*

*Figure 2.* **Rank bottlenecks leave many score / ranking / sign patterns unfeasible** (illustrated on a toy LD-KGE, $d = 1$, $|\mathcal{E}| = 3$). (a) Projection of the hidden space of queries $(s, r, ?)$ by the matrix of object vectors $\mathbf{E}$. Scores lie in a 1D linear subspace. Probabilities after softmax lie in a 1D smooth manifold of the probability simplex, leaving all other points unfeasible. (b) The object vectors $\mathbf{e}_o$ split the hidden state space into regions corresponding to different rankings / labels. Many configurations (e.g., $\boldsymbol{\pi}_{312}$ or $+-+$) are unfeasible.

## 2.2.2. OUTPUT FUNCTIONS AND TASK OBJECTIVES

The introduction of new KGEs has been accompanied by new training methods and loss functions (Ruffinelli et al., 2020; Ali et al., 2022). For a detailed overview, see Appendix D.2. While KGEs are usually evaluated on ranking metrics, the choice of the output and corresponding loss functions often reflects different underlying task objectives. We introduce three such tasks of increasing difficulty.

**Ranking reconstruction (RR)** Given a query $(s, r, ?)$, correct objects $o$ should receive a higher score than any incorrect object $o'$, i.e., $z_{s,r,o} > z_{s,r,o'}$ if $(s, r, o) \in \mathcal{G}$ and $(s, r, o') \notin \mathcal{G}$. RR is explored in margin-based loss functions (Bordes et al., 2013; Yang et al., 2014).

**Sign reconstruction (SR)** Given a query $(s, r, ?)$, $z_{s,r,o} > 0$ if $(s, r, o) \in \mathcal{G}$, and $z_{s,r,o} < 0$ otherwise. This objective can be viewed as binary classification over triples (Trouillon et al., 2017; Dettmers et al., 2018; Pezeshkpour et al., 2020).

**Distributional reconstruction (DR)** For a query $(s, r, ?)$, assign uniform high scores $z_{s,r,o} = \tau^+$ to all true triples $(s, r, o) \in \mathcal{G}$ and uniform low scores $z_{s,r,o'} = \tau^-$ to all $(s, r, o') \notin \mathcal{G}$, with $\tau^- < \tau^+$. For example, $\tau^+ = 1$ and $\tau^- = 0$ correspond to exact binary reconstruction. DR also enables probabilistic interpretations, e.g., with softmax layers defining uniform distributions over true triples. This can be relevant for downstream reasoning. For example, Arakelyan et al. (2021; 2023) combine triple probabilities to answer complex queries, Loconte et al. (2023) samples triples, and Zhu et al. (2025) uses probabilities for conformal prediction.

We will say that a KGE is *expressive* for a task under a certain condition (typically, a bound on the dimensionality $d$ of the embedding space) if there exists a parameter setting that satisfies the task objective under that condition.

## 3. Rank Bottlenecks in KGEs...

We discuss limitations of LD-KGEs in each task by showing necessary embedding dimensions which are impractical for large graphs. We define dimensional requirements in two contexts:

**Universality** Whether an LD-KGE can represent *all* possible relational patterns for *any* graph of size $|\mathcal{E}|$.

**Realisability** Whether an LD-KGE can represent patterns consistent with a *particular* graph of size $|\mathcal{E}|$.

Universality is the more common focus in KGE expressivity. However, the condition of reconstructing all graphs is both unrealistic and unnecessary in practice. While the exact ground-truth graph is unknown, realisability can give practical insights by analysing the training graph.

To develop our proofs, we consider the ground-truth adjacency matrix $\mathbf{Y} \in \{0, 1\}^{|\mathcal{E}||\mathcal{R}| \times |\mathcal{E}|}$, where $y_{i,o}$ is 1 if $o$ is an object for the subject–relation pair $i = (s, r)$ and 0 otherwise.[3] The corresponding view of the score matrix is $\mathbf{Z} = \mathbf{H}\mathbf{E}^\top \in \mathbb{R}^{|\mathcal{E}||\mathcal{R}| \times |\mathcal{E}|}$ where $\mathbf{H} = [\mathbf{h}_{s,r}]_{s,r} \in \mathbb{R}^{|\mathcal{E}||\mathcal{R}| \times d}$ stacks the hidden states of subject-relation pairs. Since the rank of $\mathbf{Z}$ is at most $d$, each score vector $\mathbf{z}_{s,r,:}$ is confined to a $d$-dimensional linear subspace of $\mathbb{R}^{|\mathcal{E}|}$. We connect these definitions with linear algebra results to derive necessary dimensions for expressing each task. Table 1 summarises the results, and Figure 2 illustrates universality bottlenecks in a toy LD-KGE with $d = 1$ and $|\mathcal{E}| = 3$.

### 3.1. ...for Solving Ranking Reconstruction

We first discuss how rank bottlenecks affect a model's ability to order potential objects, as is standard in KGE evaluation protocols. Borrowing the notation of Grivas et al. (2022), let

---

[3] We flatten subject-relations into a single axis for convenience.

*Table 1.* Necessary embedding dimensions for expressing each task objective for LD-KGEs. **The universality requirements are unrealistic, while the realisability ones depend on knowing the graph structure.** As $\mathrm{srank}(\mathbf{Y}^\pm)$ is intractable in general, we provide upper bounds in Section 4.

|  | **Rankings** | **Signs** | **Distributions** |
|---|---|---|---|
| *Universality* | $\lvert\mathcal{E}\rvert-1$ | $\lvert\mathcal{E}\rvert$ | $\lvert\mathcal{E}\rvert-1$ |
| *Realisability* | $\mathrm{srank}(\mathbf{Y}^\pm)-1$ | $\mathrm{srank}(\mathbf{Y}^\pm)$ | $\mathrm{rank}(\mathbf{Y})-1$ |

$\pi$ denote a permutation of objects. For example, in Figure 2b, the scores $\mathbf{z}_{s,r,:} = \mathbf{h}_{s,r}\mathbf{E}^\top = \begin{bmatrix} -2 & -0.5 & 3. \end{bmatrix}$ would imply that we assign the query $\mathbf{h}_{s,r}$ the ranking $\pi_{321}$ since $z_{s,r,3} > z_{s,r,2} > z_{s,r,1}$. Notice that activations like sigmoid and softmax preserve the ranking.

**Theorem 3.1** (RR universality). *The embedding dimension $d$ required for LD-KGEs to represent all possible rankings on graphs with $\lvert\mathcal{E}\rvert$ entities is at least $d \geq \lvert\mathcal{E}\rvert - 1$.*

All proofs are in Appendix B. In Figure 2b, we see that only the rankings $\pi_{123}$ and $\pi_{321}$ are feasible. If one query $(s, r, ?)$ has a different ground-truth ranking (e.g., $\pi_{213}$), then even the most powerful encoder producing $\mathbf{h}_{s,r}$ cannot achieve that ranking. This condition is a lower bound on the universal expressivity conditions of all LD-KGEs, such as the ones shown for COMPLEX ($d = \lvert\mathcal{E}\rvert\lvert\mathcal{R}\rvert$) (Trouillon et al., 2017) or TUCKER ($d = \lvert\mathcal{E}\rvert$) (Balazevic et al., 2019). In practice, $d \ll \lvert\mathcal{E}\rvert$ by several orders of magnitude, making strict universality impractical in large-scale KGs.

Let $\mathbf{Y}^\pm = 2\mathbf{Y} - 1 \in \{-1, +1\}^{\lvert\mathcal{E}\rvert\lvert\mathcal{R}\rvert \times \lvert\mathcal{E}\rvert}$ be the sign matrix corresponding to $\mathbf{Y}$, with $0$ mapped to $-1$. To show the realisability of a graph $\mathbf{Y}$, we use the *sign rank* of $\mathbf{Y}^\pm$, which is the minimum rank of any real matrix $\mathbf{M}$ with $\mathrm{sign}(\mathbf{M}) = \mathbf{Y}^\pm$. We denote it $\mathrm{srank}(\mathbf{Y}^\pm)$.

**Theorem 3.2** (RR realisability). *The embedding dimension $d$ required for LD-KGEs to realise rankings that are consistent with a graph $\mathbf{Y}$ is at least $d \geq \mathrm{srank}(\mathbf{Y}^\pm) - 1$.*

This result is a lower bound on the conditions for realisability in any particular LD-KGE. For example, Wang et al. (2018) show an upper bound to the rank required for RESCAL or COMPLEX which, as we show in Appendix E.2, is up to $2\lvert\mathcal{R}\rvert$ times larger than the one of Theorem 3.2.

The sign rank of a matrix is usually much smaller than its rank, which explains why it is easier to reconstruct rankings than distributions. Still, it cannot be evaluated in practice because determining the sign rank of a matrix is NP-hard (Alon et al., 1985). The usual approach is to bound the sign rank with other quantities that are easier to compute. Section 4 details such an upper bound using the KG connectivity.

### 3.2. ...for Solving Sign Reconstruction

We obtain a similar result for the SR task. Let $\mathbf{y} \in \{+, -\}^{\lvert\mathcal{E}\rvert}$ be the multi-label assignment of a query. For example, in Figure 2b, if we obtain the scores $\mathbf{z}_{s,r,:} = \mathbf{h}_{s,r}\mathbf{E}^\top = \begin{bmatrix} -2 & -0.5 & 3. \end{bmatrix}$, we assign the query $\mathbf{h}_{s,r}$ the multi-label assignment $\mathbf{y} = \begin{bmatrix} - & - & + \end{bmatrix}$.

**Theorem 3.3** (SR universality). *The embedding dimension $d$ required for LD-KGEs to represent all possible multi-label assignments on graphs with $\lvert\mathcal{E}\rvert$ entities is at least $d \geq \lvert\mathcal{E}\rvert$.*

For example, in Figure 2b, we see that only the multi-label assignments $++-$ and $--+$ are feasible. For any other assignment $\mathbf{y}'$, it is impossible to learn an embedding $\mathbf{h}$ such that the LD-KGE classifies it correctly.

The result for realising sign reconstructions follows directly from the definition of sign rank. We again refer to Section 4 for a useful upper bound on this requirement.

**Theorem 3.4** (SR realisability). *The embedding dimension $d$ required for LD-KGEs to realise the exact multi-label assignments for a graph $\mathbf{Y}$ is at least $d \geq \mathrm{srank}(\mathbf{Y}^\pm)$.*

### 3.3. ...for Solving Distributional Reconstruction

Representing all possible score vectors in $\mathbb{R}^{\lvert\mathcal{E}\rvert}$ naturally requires $d \geq \lvert\mathcal{E}\rvert$, while representing all probability distributions over the probability simplex $\Delta^{\lvert\mathcal{E}\rvert-1}$ requires $d \geq \lvert\mathcal{E}\rvert - 1$. However, DR only requires representing a restricted family of uniform vectors with score $\tau^+$ for all $o \in S$ and $\tau^-$ otherwise, where $S \subseteq \mathcal{E}$ are query-specific sets of objects. We confirm next that (i) this restricted family of $\tau^+, \tau^-$ score patterns still spans all of $\mathbb{R}^{\lvert\mathcal{E}\rvert}$, requiring $d \geq \lvert\mathcal{E}\rvert$, (ii) under a relaxed notion of DR where the high/low thresholds $\tau_S^+, \tau_S^-$ can be different for each set, the requirement becomes $d \geq \lvert\mathcal{E}\rvert - 1$.

**Theorem 3.5** (DR universality). *The embedding dimension $d$ required to represent all distributional reconstruction patterns with fixed global thresholds $\tau^+, \tau^-$ on graphs with $\lvert\mathcal{E}\rvert$ entities is at least $d \geq \lvert\mathcal{E}\rvert$. If distributional reconstruction is required only up to an arbitrary offset (i.e., allowing support-specific thresholds), then the requirement is $d \geq \lvert\mathcal{E}\rvert - 1$.*

The second notion allows reconstruction up to an arbitrary offset, satisfying the standard for softmax distributions $d \geq \lvert\mathcal{E}\rvert - 1$. Notice that while softmax introduces non-linear interactions, it restricts predictions to a smooth $d$-dimensional manifold within the probability simplex $\Delta^{\lvert\mathcal{E}\rvert-1}$ that is rigidly constrained to certain shapes (Figure 2a) (Finlayson et al., 2024). Because softmax is log-linear, class probability ratios must be linear on a logarithmic scale. Ground truths violating this constraint cannot be represented (Yang et al., 2018). See Appendix E.1 for details.

The result for realising distributional reconstructions is the same under both threshold notions.

**Theorem 3.6** (DR realisability). *Realising the exact uniform distributions (with fixed or support-specific thresholds) for a graph* $\mathbf{Y}$ *requires embedding dimension* $d \geq \operatorname{rank}(\mathbf{Y}) - 1$.

# 4. A Sufficient Dimension for Sign and Ranking Realisability

Theorems 3.2 and 3.4 give requirements for realising rankings and sign predictions using sign ranks, but sign ranks are NP-hard to compute and cannot be evaluated in practice. In this section, we connect a classic result that upper-bounds the sign rank by the maximum number of sign changes in any row of a matrix (Alon et al., 1985) with insights from the graph factorisation literature that connect this quantity to node degrees (Chanpuriya et al., 2020), to derive a sufficient condition for realisability expressed in subject–relation pair degrees in knowledge graphs.

Let $\mathbf{Y}^{\pm} \in \{-1, +1\}^{|\mathcal{E}||\mathcal{R}| \times |\mathcal{E}|}$ be the sign matrix representing a KG with subject–relation pairs on the first axis and objects on the second. Given an ordering $\boldsymbol{\pi}$ of the objects, the number of *sign changes* of a row $\mathbf{y}_{i,:}^{\pm} \in \{-1, +1\}^{|\mathcal{E}|}$ is $\sigma_{\boldsymbol{\pi}}(\mathbf{y}_{i,:}^{\pm}) = |\{ t : y_{i,\boldsymbol{\pi}(t)}^{\pm} \neq y_{i,\boldsymbol{\pi}(t+1)}^{\pm} \}|$, i.e. the number of adjacent objects with opposite signs once reordered by $\boldsymbol{\pi}$.

The sign rank is upper-bounded by the number of sign changes through a classical construction of Alon et al. (1985), which we restate for the KG sign matrix.

**Theorem 4.1** (Alon et al. (1985)). *For any ordering* $\boldsymbol{\pi}$ *of the objects,* $\mathbf{Y}^{\pm}$ *can be exactly decomposed as* $\mathbf{Y}^{\pm} = \operatorname{sign}(\mathbf{H}\mathbf{E}^{\top})$, *where* $\operatorname{sign}$ *is applied element-wise,* $\mathbf{H} \in \mathbb{R}^{|\mathcal{E}||\mathcal{R}| \times (\kappa+1)}$, $\mathbf{E} \in \mathbb{R}^{|\mathcal{E}| \times (\kappa+1)}$, *and* $\kappa = \max_i \sigma_{\boldsymbol{\pi}}(\mathbf{y}_{i,:}^{\pm})$ *is the maximum number of sign changes over the rows of* $\mathbf{Y}^{\pm}$. *In particular,* $\operatorname{srank}(\mathbf{Y}^{\pm}) \leq \kappa + 1$.

The proof is detailed in Appendix B. The bound holds for every ordering $\boldsymbol{\pi}$, and a smaller $\kappa$ yields a tighter bound. We can bound it with the maximum out-degree of the graph.

**Corollary 4.2.** *For any KG with maximum out-degree* $c$ *across subject–relation pairs, every ordering satisfies* $\kappa \leq 2c$. *Hence there exists an LD-KGE of rank* $d_{SR} = 2c+1$ *that realises perfect rankings and signs for queries* $(s, r, ?)$.

This corresponds to the *worst-case ordering*, in which the objects of each subject-relation pair alternate signs along the row so that every connection costs two sign changes. Table 2 presents the theoretical sufficient dimensions for several KGs. In summary, more densely connected datasets are harder to fit in theory. Finding an optimal ordering refining the bound is NP-hard. In Appendix Section B, we show how to obtain a good permutation cheaply with a Reverse Cuthill-McKee (RCM) heuristic on the object co-occurrence graph. As shown in the last columns of Table 2, this tightens the bound by roughly $5\times$ across all datasets, bringing it closer to dimensions used in practice.

Two practical questions remain. First, can existing LD-KGEs, with their specific encoders, actually represent the factorisation described in Theorem 4.1? With large enough encoding layers, Neural KGEs should be able to express this factorisation as neural networks are universal approximators (the $\mathbf{h}_{s,r}$ and $\mathbf{e}_o$ embeddings are not rigidly tied, which allows to represent the embeddings for the factorisation). However, bilinear KGEs are more limited. To our knowledge, the lowest sufficient bound in bilinear KGEs is shown by Wang et al. (2018) for RESCAL and COMPLEX and is up to $2|\mathcal{R}|$ times larger than $\operatorname{srank}(\mathbf{Y}^{\pm})$ (see Appendix E.2). It is not clear where the necessary *and* sufficient $d$ lies between these two. Second, even if some LD-KGEs can technically express this solution, can it be efficiently learned through gradient-based optimisation in practice? Interpolated solutions as pictured in Figure 4 usually reside in tight subspaces of the parameter space and are difficult to converge to.

Despite these challenges, this theoretical result provides a valuable guide for dimensionality choices and for relating different dataset connectivities.

# 5. Breaking Rank Bottlenecks with Mixtures of Softmaxes

We now propose a solution to break rank bottlenecks in LD-KGEs. A simple approach is to use a larger embedding dimension $d$ to reflect graph size or, according to the bound of Theorem 4, its connectivity. However, this quickly becomes impractical for large graphs. Instead, we adapt the *Mixture of Softmaxes* (MoS) layer proposed in language modelling (Yang et al., 2018) to LD-KGEs.

Let $O$ be a random variable over the objects. LD-KGEs use the softmax to model a categorical distribution $P(O|s, r) = \operatorname{softmax}(\mathbf{h}_{s,r}\mathbf{E}^{\top}) \in \Delta^{|\mathcal{E}|-1}$, where $\Delta^{|\mathcal{E}|-1}$ is the probability simplex over the objects. $P(O|s, r)$ should align with the data distribution (labels of the corresponding triples, normalised to sum to 1). Although softmax is traditionally used to predict a single class, it is repurposed for multi-label scenarios by retrieving top-$k$ entries, for ranking likely completions (RR), and for modelling generative distributions (DR). Extensive benchmarking has shown that training LD-KGEs with softmax and CE loss yields strong results (Ruffinelli et al., 2020; Ali et al., 2022).

As shown in Sections 3.3 and 3.1, a bottlenecked KGE model with a softmax activation cannot represent all possible distributions and rankings over the objects. We propose using a KGE-MoS layer, which is a mixture of $K$ softmaxes, to break these rank bottlenecks. Specifically,

$$P(O|s, r) = \sum_{k=1}^{K} \pi_k(\mathbf{h}_{s,r}) \operatorname{softmax}(f_k(\mathbf{h}_{s,r})\mathbf{E}^{\top}) \quad (3)$$

where $\sum_{k=1}^{K} \pi_k(\mathbf{h}_{s,r}) = 1$. $\pi_k(\mathbf{h}_{s,r}) \in [0, 1]$ is the prior

*Table 2.* Sufficient embedding dimension $d_{\text{SR}}$ for exact sign reconstruction of $(s, r, ?)$ queries on different KGs including inverse relations. The *worst-case ordering* bound uses the maximum subject–relation out-degree $c$; the *RCM ordering* bound uses the maximum number of object blocks $b$ obtained after reordering entities with a RCM heuristic (Appendix B). **The dimension scales with graph connectivity rather than simply the number of entities. A good entity ordering tightens the bound by roughly $5\times$.**

| Dataset | $\|\mathcal{E}\|$ | $\|\mathcal{R}\|$ | Worst-case ordering | | | RCM ordering | | |
|---|---|---|---|---|---|---|---|---|
| | | | Out-deg. (avg) | Out-deg. (max), $c$ | $d_{\text{SR}}$ $2c+1$ | Blocks (avg) | Blocks (max), $b$ | $d_{\text{SR}}$ $2b+1$ |
| FB15k-237 | 14,541 | 237 | 3.83 | 4,364 | 8,729 | 3.3 | 592 | 1,185 |
| Hetionet | 45,158 | 24 | 21.66 | 15,036 | 30,073 | 21.7 | 3,552 | 7,105 |
| ogbl-biokg | 93,773 | 51 | 37.48 | 29,328 | 58,657 | 29.3 | 3,025 | 6,051 |
| openbiolink | 180,992 | 28 | 18.12 | 18,420 | 36,841 | 13.7 | 4,235 | 8,471 |

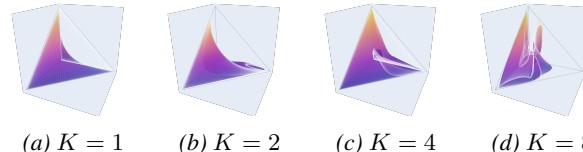

|          |          |          |          |
|:--------:|:--------:|:--------:|:--------:|
| *(a) $K = 1$* | *(b) $K = 2$* | *(c) $K = 4$* | *(d) $K = 8$* |

*Figure 3.* Feasible distributions of randomly initialised KGE-MoS output layers in a toy model ($d = 2, |\mathcal{E}| = 4$). The points are calculated for different inputs $\mathbf{h}_{s,r}$ while $\mathbf{E}$, $\boldsymbol{\omega}_k$ and $f_{\boldsymbol{\theta}_k}$ are fixed. **A higher number of softmaxes $K$ has a higher representational power and can better fit arbitrary ground-truth $P^*(O|s, r)$.**

or mixture weight of the $k$-th component and $f_k(\mathbf{h}_{s,r}) \in \mathbb{R}^d$ is the $k$-th context vector associated with the input $(s, r)$. The prior is parameterised as $\pi_k(\mathbf{h}_{s,r}) = \exp \mathbf{h}_{s,r}^\top \boldsymbol{\omega}_k / \sum_{k'} \exp \mathbf{h}_{s,r}^\top \boldsymbol{\omega}_{k'}$ with $\boldsymbol{\omega}_k \in \mathbb{R}^d$. The context vector $f_k(\mathbf{h}_{s,r})$ is obtained by mapping the hidden state with a component-specific projection with parameters $\boldsymbol{\theta}_k$. In this work, we model each $f_k$ with a two-layer MLP. We present KGE-MoS with softmaxes, which target ranking (RR) and distributional (DR) reconstruction. The same mixture also applies to sign reconstruction (SR) by replacing the softmaxes with sigmoids; we demonstrate this on `ogbl-biokg` in Appendix F.5.

Notice that only the parameters $\boldsymbol{\theta}_k$ and $\boldsymbol{\omega}_k$ are component-specific. $\mathbf{E} \in \mathbb{R}^{|\mathcal{E}| \times d}$ is shared, resulting in a low additional parameter cost of $O(Kd + K|\boldsymbol{\theta}_k|)$ – in our case, $O(Kd^2)$. In contrast, increasing the dimension of the LD-KGE by a factor $d_{\text{inc}}$ has a cost $O(d_{\text{inc}}(|\mathcal{E}| + |\mathcal{R}|))$ which is usually less scalable as $|\mathcal{E}| \gg Kd$.

**Mixtures break rank bottlenecks efficiently** Without increasing the manifold dimension, the weighted mixture allows for richer non-linear distribution representations (see Figure 3). From a matrix perspective, a single softmax ($K = 1$) restricts the log-probability matrix $\mathbf{A} \in \mathbb{R}^{|\mathcal{E}||\mathcal{R}| \times |\mathcal{E}|}$, where $\mathbf{a}_{i,:} = \log P(O|i)$, to a $d + 1$ linear subspace. In contrast, constructive evidence shows that mixing softmaxes enables a superlinear increase of ranks (Yang et al., 2018). Our empirical tests on FB15K237 (see Appendix F.4.2) confirm this: with $d = 200$ and $K = 4$, KGE-

MoS approaches full rank ($\text{rank}(\mathbf{A}) = 14,501$, not merely $Kd = 800$) while adding only $10\%$ more parameters.

**Subject prediction** KGE-MoS is a drop-in output layer that can be applied to any LD-KGE presented in Section 2.2.1. Notice that while bilinear models (e.g., DISTMULT) can perform subject prediction $(?, r, o)$ scalably as a vector-matrix multiplication, their augmentations (e.g., DISTMULT-MoS) must rely on inverse relations $(o, r^{-1}, ?)$. We discuss this in Appendix C.

# 6. Experiments

We aim to answer the following research questions: **(RQ1)** "How does the performance of models with the proposed KGE-MoS output layer compare to bottlenecked KGE models?" **(RQ2)** "How does increasing the embedding size, as a simple alternative, compare to KGE-MoS in performance?" Our code is available at https://github.com/SonyResearch/KGEMoS.

**Datasets** We evaluate LD-KGEs on the following standard link prediction datasets: FB15k-237 (Toutanova & Chen, 2015), Hetionet (Himmelstein et al., 2017), ogbl-biokg (Hu et al., 2020), and openbiolink (Breit et al., 2020). FB15k-237 is a commonly used benchmark, but its size is relatively small compared to modern KGs. We choose the other datasets for their larger sizes, where we expect rank bottlenecks to be more impactful due to our theoretical results. See Appendix F.1 for statistics.

**Models** For baseline LD-KGEs, we use DISTMULT (Yang et al., 2014), COMPLEX (Trouillon et al., 2017), RESCAL (Nickel et al., 2011) and CONVE (Dettmers et al., 2018), each trained with a softmax output layer and inverse relations for subject prediction, since they are standard choices for large-scale link prediction (Ali et al., 2022). We also include COMPGCN (Vashishth et al., 2020) to study a more expressive neural encoder. We use the best-performing hyperparameters found by Ruffinelli et al. (2020) (see Ap-

pendix F.3). We compare the baselines with their high-rank variants DISTMULT-MoS, RESCAL-MoS, etc., replacing the softmax output layer with KGE-MoS using $K = 4$ softmaxes. We train each model with embedding dimensions $d = 200$ and $d = 1000$, except for COMPLEX, where we halve the dimensions for comparability because the model has a bottleneck of $2d$ (see Appendix D.1). For COMPGCN, we only evaluate in the low-dimensional regime due to computational constraints. The hyperparameters for the KGE-MoS output layer are found via random search on `ogbl-biokg` and are detailed in Appendix F.3.

**Metrics** As usual (Nickel et al., 2015; Ruffinelli et al., 2020; Ali et al., 2022), we assess performance on object queries $(s, r, ?)$ and subject queries $(?, r, o)$ using filtered mean reciprocal rank (MRR) (see Appendix F.2). We measure distributional fidelity using the negative log likelihood (NLL) of the model predictions on the test set. Whereas previous work (Loconte et al., 2023) directly reports the NLL, we introduce a *filtered NLL* metric that more accurately reflects the predictive performance of a model for link prediction. This prevents penalising models that assign low total probability mass outside training triples. See Appendix F.2 for more details.

### 6.1. Results

Table 3 reports our results. We run each model three times per dataset for statistical significance.

**(RQ1)** **On the three larger and more densely connected datasets, the best-performing models are consistently KGE-MoS models in both low- and high-dimensional regimes**. In other words, gains are strongest exactly where theory predicts they matter. The best-performing models are often DISTMULT-MoS or COMPLEX-MoS, hinting that a simple encoder for the subject–relation with an elaborate object output layer might be the most effective approach. In contrast, on the smallest dataset FB15k-237, KGE-MoS does not improve and sometimes even hurts the performance of bottlenecked KGE models. In fact, the best LD-KGE at $d = 200$ performs nearly as well as the best LD-KGE at $d = 1000$, hinting that rank bottlenecks are not critical in small-scale datasets. We confirm this in Appendix F.4, where we report similar results on WN18RR, a dataset with even lower connectivity than FB15k-237. Performance degradation is likely due to overfitting rather than instability, as standard deviations on large, dense graphs (Appendix F.4) show little variation across runs.

**(RQ2)** Comparing models across dimensions, we find that **on the three larger datasets, KGE-MoS at $d = 200$ achieves performance close to the bottlenecked baselines at $d = 1000$**. Still, the performance increase of KGE-MoS

models is larger at $d = 1000$ than at $d = 200$, suggesting that embedding dimension remains crucial for representational power. In an additional experiment in Appendix F.4.4, we tried to evaluate LD-KGEs in larger datasets at $d = 5000$ to push the embedding dimension until bottlenecks do not matter. We find that **simply increasing embedding size past some high value requires reducing the training batch size and hurts performance.** This contrasts with the parameter efficiency of KGE-MoS.

$K$ **ablation** Next, we run an ablation on the number of softmaxes $K$ used in KGE-MoS. We use `ogbl-biokg` which Section 4 suggests to be challenging due to its high average connectivity. We analyse DISTMULT-MoS and COMPLEX-MoS, the best-performing models. Table 4 details the results. Each value was averaged over three runs (see Table 16 in Appendix for standard deviations). Increasing the number of softmaxes consistently leads to better performance. To confirm that the improvements come from the mixture and not merely the additional encoding power of the projection $f_{\theta_k}$, we also evaluate the MoS models at $K = 1$, which use $f_{\theta_k}$ without mixing. These models outperform the original models while still bottlenecked, but not as much as those that break the bottleneck with $K > 1$. Appendix F.4.5 further ablates the projection depth, showing that a more expressive $f_{\theta_k}$ (two layers vs. one) improves performance on top of the gains from mixing.

**Computational cost** While KGE-MoS has a marginal parameter cost, its computational cost is higher than that of a regular softmax layer. On `openbiolink`, the dataset with the largest output layer, we find that KGE-MoS at $K = 4$ is between 1.69 and 2.75 times slower than its bottlenecked baseline for a training step. The largest difference is recorded for DISTMULT and DISTMULT-MoS, since DISTMULT is a very simple model. Still, we notice that the number of softmaxes has a negligible impact on inference time. See Table 8 for details. We find that most of the overhead is due to the computation of the projections $f_{\theta_k}(\mathbf{h}_{s,r})$. Therefore, if inference time is a concern, this can be mitigated by using a more efficient projection layer.

## 7. Related work

**Rank bottlenecks** Rank bottlenecks in output layers of neural networks were first studied in *softmax models* for language modelling (Yang et al., 2018). If the task is to predict a single best completion for a query, low-rank constraints are less problematic (Grivas et al., 2022). However, to predict a distribution over all plausible completions, rank bottlenecks severely limit the expressivity of the model. Yang et al. (2018) proposed a mixture of softmaxes as an output layer to break the bottleneck in language modelling. Yang et al. (2019) proposed a variant, more scalable mixture model,

*Table 3.* **KGE-MoS improves performance and probabilistic fit of LD-KGEs on the three larger-scale datasets**, where theory predicts bottlenecks matter. Average NLL ↓ and MRR ↑. Standard deviations and Hits@ metrics are reported in Appendix F.4. Best results per dataset and dimension are in **bold**. We also report the average improvement in NLL and MRR (mixture vs. regular). ⋆ indicates statistical significance at $p < 0.05$ in a Wilcoxon signed-rank test pairing each model with its mixture variant.

| MODEL | FB15K-237 | | | HETIONET | | | OGBL-BIOKG | | | OPENBIOLINK | | |
|---|---|---|---|---|---|---|---|---|---|---|---|---|
| | NLL | MRR | PARAM | NLL | MRR | PARAM | NLL | MRR | PARAM | NLL | MRR | PARAM |
| *d = 200* | | | | | | | | | | | | |
| DISTMULT | 4.74 | .304 | 3.0M | 6.10 | .250 | 9.0M | 4.83 | .792 | 18.8M | 5.14 | .302 | 36.2M |
| DISTMULT-MoS | 4.65 | .306 | 3.3M | **5.83** | **.277** | 9.4M | 4.65 | .792 | 19.1M | **5.03** | .314 | 36.5M |
| COMPLEX [†] | 4.74 | .303 | 3.0M | 6.10 | .249 | 9.0M | 4.83 | .792 | 18.8M | 5.13 | .301 | 36.2M |
| COMPLEX-MoS [†] | 4.71 | .301 | 3.3M | 5.85 | .269 | 9.4M | **4.65** | **.793** | 19.1M | 5.06 | .313 | 36.5M |
| RESCAL | 4.79 | .258 | 21.9M | 6.13 | .219 | 10.9M | 4.89 | .763 | 22.8M | 5.16 | .303 | 38.4M |
| RESCAL-MoS | 4.65 | .318 | 22.2M | 5.87 | .274 | 11.3M | 4.70 | .780 | 23.2M | 5.04 | **.323** | 38.8M |
| CONVE | **4.48** | **.321** | 5.1M | 6.03 | .252 | 11.1M | 4.94 | .782 | 20.8M | 5.28 | .286 | 38.3M |
| CONVE-MoS | 4.57 | .311 | 5.4M | 5.92 | .263 | 11.4M | 4.77 | .768 | 21.2M | 5.10 | .304 | 38.6M |
| COMPGCN [‡] | 4.80 | .300 | 1.6M | 5.95 | .260 | 4.7M | 5.11 | .749 | 9.5M | 5.43 | .263 | 18.3M |
| COMPGCN-MoS [‡] | 4.86 | .310 | 2.0M | 5.86 | .266 | 5.0M | 4.96 | .744 | 9.6M | 5.43 | .274 | 18.3M |
| *avg -MoS delta* | -.02 | .006 | | ⋆-.19 | ⋆.022 | | ⋆-.17 | .000 | | ⋆-.12 | ⋆.015 | |
| *d = 1000* | | | | | | | | | | | | |
| DISTMULT | **4.56** | **.331** | 15.0M | 6.04 | .288 | 45.2M | 4.89 | .801 | 93.9M | 5.17 | .316 | 181.0M |
| DISTMULT-MoS | 4.72 | .311 | 23.0M | 5.76 | .312 | 53.2M | **4.34** | **.837** | 101.9M | **4.89** | **.347** | 189.0M |
| COMPLEX [†] | 4.71 | .317 | 15.0M | 5.99 | .292 | 45.2M | 4.86 | .806 | 93.9M | 5.12 | .322 | 181.0M |
| COMPLEX-MoS [†] | 4.64 | .314 | 23.0M | 5.78 | .303 | 53.2M | 4.39 | .836 | 101.9M | 4.87 | .345 | 189.0M |
| RESCAL | 4.64 | .307 | 488.5M | 5.93 | .243 | 93.1M | 4.74 | .799 | 195.8M | 5.03 | .328 | 237.0M |
| RESCAL-MoS | 4.63 | .325 | 496.5M | 5.87 | .300 | 101.2M | 4.42 | .824 | 203.8M | 5.00 | .328 | 245.0M |
| CONVE | 4.65 | .301 | 70.1M | 6.06 | .262 | 100.3M | 4.93 | .807 | 149.0M | 5.24 | .308 | 236.2M |
| CONVE-MoS | 4.69 | .316 | 78.2M | **5.71** | **.313** | 108.3M | 4.43 | .817 | 157.0M | 4.91 | .336 | 244.2M |
| *avg -MoS delta* | .02 | .002 | | ⋆-.23 | ⋆.035 | | ⋆-.43 | ⋆.024 | | ⋆-.22 | ⋆.021 | |

[†] Results for COMPLEX use halved $d = 100$ and $d = 500$. It has real and imaginary parameters, leading to a rank bottleneck of $2d$.

[‡] Results for COMPGCN use $d = 100$ on ogbl-biokg and $d = 50$ on openbiolink due to computational constraints.

*Table 4.* MRR and NLL on ogbl-biokg at $d = 1000$ with different numbers of softmaxes.

| MODEL | #SOFTMAX | NLL ↓ | MRR ↑ | MODEL | #SOFTMAX | NLL ↓ | MRR ↑ |
|---|---|---|---|---|---|---|---|
| DISTMULT | 1 | 4.89 | .801 | COMPLEX | 1 | 4.86 | .806 |
| DISTMULT-MoS | 1 | 4.42 | .821 | COMPLEX-MoS | 1 | 4.46 | .820 |
| DISTMULT-MoS | 2 | 4.37 | .831 | COMPLEX-MoS | 2 | 4.41 | .827 |
| DISTMULT-MoS | 4 | 4.34 | .837 | COMPLEX-MoS | 4 | 4.39 | .836 |
| DISTMULT-MoS | 8 | **4.33** | **.841** | COMPLEX-MoS | 8 | **4.37** | **.838** |

whereas Kanai et al. (2018); Ganea et al. (2019) proposed alternative non-linearities. Grivas et al. (2024) explored rank bottlenecks in clinical and image *multi-label classification* and proposed a discrete Fourier transform layer to guarantee a minimum number of label combinations to be feasible. Finally, Weller et al. (2026) explored rank bottlenecks in information retrieval for document ranking.

**Expressivity of LD-KGEs** Earlier works (Trouillon et al., 2017; Wang et al., 2018; Kazemi & Poole, 2018; Balazevic et al., 2019) have given theoretical guarantees on the expressivity of specific bilinear KGEs for ranking reconstruction. However, the provided conditions (e.g., $d \geq |\mathcal{E}|$) are unrealistic for large datasets, and our work is the first to analyse this issue empirically for large and dense graphs. To the best of our knowledge, we are also the first to provide bounds that apply to any LD-KGE, including bilinear and neural KGEs. We also explore expressivity in other prediction tasks previously not considered in discussions, in light

of recent work (Arakelyan et al., 2021; 2023; Harzli et al., 2023; Gregucci et al., 2025; Zhu et al., 2025) which has emphasised the importance and lack of well-distributed KGE scores for downstream reasoning. KGE-MoS is reminiscent of ensemble-based KGEs (Wang et al., 2018). However, ensembles (i) are still bottlenecked as they combine models linearly, and (ii) have a high parameter cost. See Appendix G for more details.

**Other KGE paradigms** Next, we discuss alternative approaches to link prediction other than LD-KGEs as described in Section 2.2.1, which are outside the scope of this work. *Translation- and rotation-based KGEs* (Bordes et al., 2013; Wang et al., 2014; Lin et al., 2015; Sun et al., 2019) score objects using distance metrics like $\|\mathbf{h}_{s,r} - \mathbf{e}_o\|$, where relations act as geometric transformations in vector space. Translation models are interpretable and fast, but are generally not fully expressive (Kazemi & Poole, 2018; Abboud et al., 2020). On the one expressivity result we are aware of

(SR universality), switching to a Euclidean score geometry does not improve over LD-KGEs. We discuss this bound in Appendix F.4.6 and show empirically that translation models perform comparably to the bottlenecked baselines on `ogbl-biokg`. *Region-based KGEs* (Abboud et al., 2020; Pavlović & Sallinger, 2023) replace vector translations with geometric containment. Entities are embedded as points and relations as boxes. Scoring is based on whether objects lie within the box defined by the subject and relation. These KGEs prioritise spatial interpretability and complex inference patterns (e.g., hierarchical rule injection). They can be shown to be fully expressive, but with large dimension bounds – e.g., $d \geq |\mathcal{E}||\mathcal{R}|$ (Pavlović & Sallinger, 2023) – which are impractical for large datasets. Their performance in a low-dimensional regime should be studied further to investigate other forms of bottlenecks. Finally, some models for link prediction do not use a traditional KGE-based scoring function, but have more complex reasoning mechanisms like *graph sampling* (Bi et al., 2023) or *path-based reasoning* (Das et al., 2018) like NBFNet (Zhu et al., 2021), which are not subject to our analysis. Unlike the models studied in our paper, NBFNet non-linearly encodes subject and object jointly and works well in recent tests. However, it is challenging to scale to large datasets. In fact, we show in Appendix F.4.7 that even a simple non-linear decoder – an MLP applied to the concatenated embeddings of subject, relation, and object – incurs a compute and memory overhead when scoring all objects that does not scale to large knowledge graphs, unlike the drop-in KGE-MoS layer.

## 8. Conclusions

In this paper, we showed that rank bottlenecks are a fundamental limitation of many standard KGE models. We demonstrated how these bottlenecks limit model expressivity, affecting ranking accuracy and probabilistic fidelity. To address this, we introduced KGE-MoS, a mixture-based output layer that breaks rank bottlenecks efficiently. Our experiments show that KGE-MoS improves the performance of linear-decoder KGEs on large KGs at a low parameter cost. Our findings suggest that exploring non-linear output layers is a promising avenue for advancing KGEs.

## Impact Statement

This paper presents work whose goal is to advance the field of Machine Learning. There are many potential societal consequences of our work, none which we feel must be specifically highlighted here.

## Acknowledgements

We would like to express our gratitude to Andreas Grivas, Daniel Daza, Pablo Sanchez Martin, Samuel Cognolato, Tarek R. Besold, Antonio Vergari, and Pasquale Minervini for fruitful discussions during the writing of this paper. Emile van Krieken was funded by the NWO AiNed project "Human-Centric AI Agents with Common Sense" (NGF.1607.22.044). Luciano Serafini was funded by the NRRP project Future AI Research (FAIR - PE00000013) under the NRRP MUR program, funded by NextGenerationEU.

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

## A. Tensor Notation

We borrow the notation from (Cichocki et al., 2015) for tensors and matrices, summarised in Table 5.

*Table 5.* The tensor notation used throughout the paper.

| Notation | Description |
|----------|-------------|
| $\mathcal{A}, \mathbf{A}, \mathbf{a}, a$ | Tensor, matrix, vector, scalar. |
| $\mathbf{A} = \begin{bmatrix} \mathbf{a}_1 & \dots & \mathbf{a}_R \end{bmatrix}$ | Matrix $\mathbf{A}$ with columns $\mathbf{a}_r$. |
| $\mathbf{A} = \begin{bmatrix} \mathbf{a}_1 \\ \vdots \\ \mathbf{a}_R \end{bmatrix}$ | Matrix $\mathbf{A}$ with rows $\mathbf{a}_r$. |
| $a_{i_1,i_2,i_3,\dots,i_N}$ | Scalar of tensor $\mathcal{A}$ obtained by fixing all indices. |
| $\mathbf{a}_{:,i_2,i_3,\dots,i_N}$ | Fiber of tensor $\mathcal{A}$ obtained by fixing all but one index. |
| $\mathbf{A}_{:,:,i_3,\dots,i_N}$ | Matrix slice of tensor $\mathcal{A}$ obtained by fixing all but two indices. |
| $\mathcal{A}_{:,:,:,i_4,\dots,i_N}$ | Tensor slice of tensor $\mathcal{A}$ obtained by fixing some indices. |

## B. Proofs

**Theorem 3.1** (RR universality). *The embedding dimension $d$ required for LD-KGEs to represent all possible rankings on graphs with $|\mathcal{E}|$ entities is at least $d \geq |\mathcal{E}| - 1$.*

*Proof.* We use a result from Cover (1967); Good & Tideman (1977); Grivas et al. (2022) which states that a linear (affine) classifier over $N$ classes parameterised by a weight matrix $\mathbf{W} \in \mathbb{R}^{N \times d}$ (and bias vector $\mathbf{b} \in \mathbb{R}^N$) followed by any monotonic activation, can only predict a subset of class rankings if $d < N-1$. $N-1$ suffices because, in the context of ranking, a constant shift to all scores in the direction of $\mathbf{1}$ does not change the ranking. We are therefore operating in the quotient space $\mathbb{R}^N / \mathrm{span}(\mathbf{1})$ which has dimension $N - 1$. Replacing $\mathbf{W} \in \mathbb{R}^{N \times d}$ with $\mathbf{E} \in \mathbb{R}^{|\mathcal{E}| \times d}$ yields the theorem.

Note that, for object vectors in general position, if we fix $d$, changing $\mathbf{E}$ changes the set of rankings that are feasible, but not the cardinality of the set. The cardinality can be calculated using Stirling numbers with a proof that stems from finding the maximum and generic number of distance-based orderings of $N$ points in a $d$-dimensional space (Cover, 1967; Good & Tideman, 1977; Smith, 2014). The narrower the low-rank $d$, the more class permutations $\boldsymbol{\pi}$ become infeasible. $\square$

**Theorem 3.2** (RR realisability). *The embedding dimension $d$ required for LD-KGEs to realise rankings that are consistent with a graph $\mathbf{Y}$ is at least $d \geq \mathrm{srank}(\mathbf{Y}^{\pm}) - 1$.*

*Proof.* Let $\mathbf{Z}^{\star}$ be a matrix of model scores that achieves

a perfect ranking for every subject-relation pair. For each subject-relation row $i$, there is a threshold $\tau_i \in \mathbb{R}$ such that all true objects have a score strictly greater than $\tau_i$. In other words, $\text{sign}(\mathbf{z}_{i,:}^\star - \tau_i \mathbf{1}) = \mathbf{y}_{i,:}^\pm$ where $\mathbf{1}$ is the all-ones row vector. Stacking the row thresholds into a matrix $\mathbf{T} = \begin{bmatrix} \tau_1 \\ \vdots \\ \tau_{|\mathcal{E}||\mathcal{R}|} \end{bmatrix} \mathbf{1}^\top$, we get $\text{sign}(\mathbf{Z}^\star - \mathbf{T}) = \mathbf{Y}^\pm$. Since $\mathbf{Z}^\star$ has rank at most $d$ and $\mathbf{T}$ has rank 1, $\mathbf{Z}^\star - \mathbf{T}$ has rank at most $d + 1$. However, by definition of the sign rank, any matrix $\mathbf{M}$ with $\text{sign}(\mathbf{M}) = \mathbf{Y}^\pm$ satisfies $\text{rank}(\mathbf{M}) \geq \text{srank}(\mathbf{Y}^\pm)$. Therefore, the result is only possible if $d+1 \geq \text{srank}(\mathbf{Y}^\pm)$. $\square$

**Theorem 3.3** (SR universality). *The embedding dimension $d$ required for LD-KGEs to represent all possible multi-label assignments on graphs with $|\mathcal{E}|$ entities is at least $d \geq |\mathcal{E}|$.*

*Proof.* We use the result from Cover (1965); Grivas et al. (2024) which states that a linear (affine) classifier over $N$ classes parameterised by a low-rank weight matrix $\mathbf{W} \in \mathbb{R}^{N \times d}$ (and bias vector $\mathbf{b} \in \mathbb{R}^N$) can only predict a subset of multi-label assignments configurations $\mathbf{y}$ if $d < N$ using the decision rule $\text{sign}(\mathbf{W}\mathbf{x} + \mathbf{b})$ for input $\mathbf{x} \in \mathbb{R}^d$. Replacing $\mathbf{W} \in \mathbb{R}^{N \times d}$ with $\mathbf{E} \in \mathbb{R}^{|\mathcal{E}| \times d}$ yields the theorem. For a linear layer, the number of feasible assignments is upper-bounded by $2 \sum_{i=0}^{d-1} \binom{N-1}{i}$. For an affine layer, more assignments are feasible, but the increase in the number of feasible assignments is less than what would be achieved by increasing the embedding dimension $d$ by one. $\square$

**Theorem 3.4** (SR realisability). *The embedding dimension $d$ required for LD-KGEs to realise the exact multi-label assignments for a graph $\mathbf{Y}$ is at least $d \geq \text{srank}(\mathbf{Y}^\pm)$.*

*Proof.* Directly follows from the definition of the sign rank. $\square$

**Theorem 3.5** (DR universality). *The embedding dimension $d$ required to represent all distributional reconstruction patterns with fixed global thresholds $\tau^+, \tau^-$ on graphs with $|\mathcal{E}|$ entities is at least $d \geq |\mathcal{E}|$. If distributional reconstruction is required only up to an arbitrary offset (i.e., allowing support-specific thresholds), then the requirement is $d \geq |\mathcal{E}| - 1$.*

*Proof.* **Fixed thresholds.** We show that the span of the score vectors to represent is the span of the standard basis vectors of $\mathbb{R}^n$

Let $|\mathcal{E}| = n$. For any support $S \subseteq \{1, \dots, n\}$, let $\mathbf{1}_S \in \{0,1\}^n$ denote its indicator vector. That is, $(\mathbf{1}_S)_i = 1$ if $i \in S$ and 0 otherwise. Under DR with fixed global thresholds $\tau^+ > \tau^-$, the model must output the exact score vectors

$$\mathbf{z}_S = (\tau^+ - \tau^-)\mathbf{1}_S + \tau^- \mathbf{1}$$

for all supports $S$. This yields the inclusion $\text{span}\{\mathbf{z}_S : S \subseteq [n]\} \subseteq \text{span}\{\mathbf{1}_S : S \subseteq [n]\} + \text{span}\{\mathbf{1}\}$, where

$[n]$ denotes the set $\{1, \dots, n\}$. Since $\mathbf{1} = \mathbf{1}_{[n]}$, we have $\text{span}\{\mathbf{1}_S : S \subseteq [n]\} \supseteq \text{span}\{\mathbf{1}\}$. Therefore, we have simply $\text{span}\{\mathbf{z}_S : S \subseteq [n]\} \subseteq \text{span}\{\mathbf{1}_S : S \subseteq [n]\}$.

To obtain the reverse inclusion, we write the equality

$$\mathbf{1}_S = \frac{1}{\tau^+ - \tau^-}(\mathbf{z}_S - \tau^- \mathbf{1})$$

which gives $\text{span}\{\mathbf{1}_S : S \subseteq [n]\} \subseteq \text{span}\{\mathbf{z}_S : S \subseteq [n]\} + \text{span}\{\mathbf{1}\}$. Notice that $\mathbf{1} \propto \mathbf{z}_{[n]} = \tau^+ \mathbf{1}$ and therefore $\text{span}\{\mathbf{z}_S : S \subseteq [n]\} \supseteq \text{span}\{\mathbf{1}\}$. This leads to the reverse inclusion $\text{span}\{\mathbf{1}_S : S \subseteq [n]\} \subseteq \text{span}\{\mathbf{z}_S : S \subseteq [n]\}$.

Putting the two inclusions together, we get $\text{span}\{\mathbf{z}_S : S \subseteq [n]\} = \text{span}\{\mathbf{1}_S : S \subseteq [n]\}$. And since the singleton indicators $\mathbf{1}_{\{i\}}$ are the standard basis vectors, $\text{span}(\{\mathbf{1}_S : S \subseteq [n]\}) = \mathbb{R}^n$. So, to span the space of all $\mathbf{z}_S$, the bottleneck dimension must satisfy $d \geq n$.

**Support-specific thresholds.** In this setting, the thresholds $\tau_S^+, \tau_S^-$ may vary per support. For any support $S$ and its corresponding indicator vector $\mathbf{1}_S$, the target score vector is valid if it matches $\mathbf{1}_S$ up to an arbitrary affine transformation (scaling and shifting)

$$\mathbf{z}_S' = \alpha_S \mathbf{1}_S + \beta_S \mathbf{1}$$

where $\alpha_S \neq 0$ is the support-specific margin and $\beta_S$ is the offset. This means that the model is only required to produce $\mathbf{z}_S$ (from the fixed thresholds case) modulo adding a scalar multiple of the all-ones vector $\mathbf{1}$. As such, we work in the quotient space $\mathbb{R}^n / \text{span}(\mathbf{1})$ which has dimension $n - 1$ (same as for softmax). Therefore it suffices that $d \geq n - 1$. $\square$

**Theorem 3.6** (DR realisability). *Realising the exact uniform distributions (with fixed or support-specific thresholds) for a graph $\mathbf{Y}$ requires embedding dimension $d \geq \text{rank}(\mathbf{Y}) - 1$.*

*Proof.* **Fixed thresholds.** Consider a perfect solution $\mathbf{Z}^\star$ which assigns a uniform high score $\tau_+$ to true triples and a uniform low score $\tau_-$ to false ones. The solution should satisfy $\mathbf{Y} = \frac{1}{\tau_+ - \tau_-}(\mathbf{Z}^\star - \tau_- \mathbf{1})$, where $\mathbf{1}$ is a matrix of ones with the same size as $\mathbf{Y}$. Since $\mathbf{Z}^\star$ has rank at most $d$ and $\tau_- \mathbf{1}$ is a matrix of rank at most 1, the equality is only possible if $\text{rank}(\mathbf{Y}) \leq d + 1$.

**Support-specific thresholds.** Each score vector $\mathbf{z}_{s,r,:}^\star$ must be constant on the true object set $S$ and constant on its complement, but the two constants may depend on $(s, r)$. That is, $z_{i,o}^\star = \tau_i^+$ if $y_{i,o} = 1$ and $z_{i,o}^\star = \tau_i^-$ if $y_{i,o} = 0$ where $\tau_i^+ > \tau_i^-$ are row-dependent scalars. For each row $i$, subtracting the (row-dependent) offset $\tau_i^-$ gives the relative form

$$z_{i,o}^\star - \tau_i^- = (\tau_i^+ - \tau_i^-)y_{i,o}.$$

Let $\mathbf{t}^-$ denote the vector of $\tau_i^-$ values and $\mathbf{t}^-\mathbf{1}^\top$ the matrix whose i-th row equals $\tau_i^-\mathbf{1}^\top$. The above equality in matrix form is

$$\mathbf{Z}^\star - \mathbf{t}^-\mathbf{1}^\top = \operatorname{diag}(\tau_i^+ - \tau_i^-)\,\mathbf{Y}.$$

The matrix on the left has rank at most $d+1$, because $\mathbf{Z}^\star$ has rank at most $d$ and $\mathbf{t}^-\mathbf{1}^\top$ has rank 1. The matrix on the right is obtained from $\mathbf{Y}$ by left-multiplying with a diagonal matrix, which does not change its rank. Therefore, $\operatorname{rank}(\mathbf{Y}) \leq d+1$. $\qquad\square$

Recall from Section 4 that $\mathbf{Y}^\pm \in \{-1,+1\}^{|\mathcal{E}||\mathcal{R}|\times|\mathcal{E}|}$ is the KG sign matrix, that $\sigma_{\boldsymbol{\pi}}(\mathbf{y}_{i,:}^\pm)$ is the number of sign changes of its i-th row under an ordering $\boldsymbol{\pi}$ of the objects, and that $\kappa = \max_i \sigma_{\boldsymbol{\pi}}(\mathbf{y}_{i,:}^\pm)$.

**Theorem 4.1** (Alon et al. (1985))**.** *For any ordering $\boldsymbol{\pi}$ of the objects, $\mathbf{Y}^\pm$ can be exactly decomposed as $\mathbf{Y}^\pm = \operatorname{sign}(\mathbf{H}\mathbf{E}^\top)$, where* sign *is applied element-wise, $\mathbf{H} \in \mathbb{R}^{|\mathcal{E}||\mathcal{R}|\times(\kappa+1)}$, $\mathbf{E} \in \mathbb{R}^{|\mathcal{E}|\times(\kappa+1)}$, and $\kappa = \max_i \sigma_{\boldsymbol{\pi}}(\mathbf{y}_{i,:}^\pm)$ is the maximum number of sign changes over the rows of $\mathbf{Y}^\pm$. In particular, $\operatorname{srank}(\mathbf{Y}^\pm) \leq \kappa + 1$.*

*Proof.* We restate the construction of Alon et al. (1985) (applied to graphs by Chanpuriya et al., 2020), instantiated for the rectangular KG sign matrix $\mathbf{Y}^\pm \in \{-1,+1\}^{|\mathcal{E}||\mathcal{R}|\times|\mathcal{E}|}$, whose $|\mathcal{E}||\mathcal{R}|$ rows are indexed by subject–relation pairs (the *source nodes*) and whose $|\mathcal{E}|$ columns are indexed by objects (the *destination nodes*). Unlike the usual boolean adjacency matrices, we use $+1$ (or simply $+$) to denote the existence of an edge and $-1$ (or simply $-$) to denote its absence. $c$ denotes the maximum out-degree across subject–relation pairs.

Fix an ordering $\boldsymbol{\pi}$ of the objects, i.e. a bijection placing each object at a position $t \in \{1,\dots,|\mathcal{E}|\}$. For a row $\mathbf{y}_{i,:}^\pm \in \{-1,+1\}^{|\mathcal{E}|}$, let $\kappa_i = \sigma_{\boldsymbol{\pi}}(\mathbf{y}_{i,:}^\pm)$ be its number of sign changes under $\boldsymbol{\pi}$ – the number of adjacent positions $t, t+1$ carrying opposite signs – and let $\kappa = \max_i \kappa_i$ be the maximum across all subject–relation pairs.

Our goal is to decompose $\mathbf{Y}^\pm$ by expressing each row $\mathbf{y}_{i,:}^\pm$ through the equation

$$\operatorname{sign}(\mathbf{E}\mathbf{h}_i) = \mathbf{y}_{i,:}^\pm,$$

where

- sign is the sign function applied element-wise,

- $\mathbf{E} \in \mathbb{R}^{|\mathcal{E}|\times(\kappa+1)}$ is a Vandermonde matrix – that is, a matrix with $\kappa + 1$ geometric progressions for $|\mathcal{E}|$ variables – with one row per object, and

- $\mathbf{h}_i \in \mathbb{R}^{\kappa+1}$ is a vector of polynomial coefficients associated with the subject–relation pair $i$.

That is, $\mathbf{h}_i$ defines the coefficients of a polynomial of degree at most $\kappa$ which, evaluated at each object $t = 1,\dots,|\mathcal{E}|$, yields a score $p_i(t)$ that determines the sign of the edge between pair $i$ and object $t$.

If we successfully find such coefficients $\mathbf{h}_i$ for each subject–relation pair $i = 1,\dots,|\mathcal{E}||\mathcal{R}|$ – which will be guaranteed by the degree $\kappa$ of the polynomials –, then we can reconstruct the entire sign matrix $\mathbf{Y}^\pm$ as

$$\mathbf{Y}^\pm = \operatorname{sign}(\mathbf{H}\mathbf{E}^\top),$$

where $\mathbf{H} \in \mathbb{R}^{|\mathcal{E}||\mathcal{R}|\times(\kappa+1)}$ is the matrix of coefficients $\mathbf{h}_i$ for all subject–relation pairs stacked as rows.

**Constructing the Vandermonde Matrix** We define the Vandermonde matrix $\mathbf{E}$ such that

$$e_{t,j} = t^{j-1}, \quad t = 1,\dots,|\mathcal{E}|,\ j = 1,\dots,\kappa+1,$$

where $t$ indexes the objects in the order given by $\boldsymbol{\pi}$. That is, evaluating $\mathbf{E}\mathbf{h}_i \in \mathbb{R}^{|\mathcal{E}|}$ means evaluating the degree-$\kappa$ polynomial $p_i(t)$ at the integers $t = 1,\dots,|\mathcal{E}|$, each integer $t$ representing the object placed at position $t$ by $\boldsymbol{\pi}$.

**Fitting the Polynomials** Next, we wish to construct the coefficients $\mathbf{h}_i$ such that the polynomial $p_i(t)$ matches the sign pattern of $\mathbf{y}_{i,:}^\pm$:

- $p_i(t) > 0$ for indices where $y_{i,t}^\pm = +1$ and

- $p_i(t) < 0$ for indices where $y_{i,t}^\pm = -1$.

To achieve this, we place a root of $p_i(t)$ in each interval between two adjacent positions of opposite sign. A row with $\kappa_i$ sign changes thus needs exactly $\kappa_i$ roots, so a polynomial of degree $\kappa_i \leq \kappa$ suffices.

Consider for example the row $\mathbf{y}_{i,:}^\pm = \begin{bmatrix} - & - & + & - & + & - \end{bmatrix}$, which has $\kappa_i = 4$ sign changes. We need $p_i(t)$ to be positive for $t = 3$ and $t = 5$ and negative for all other $t$. We can choose $p_i(t)$ to have two roots at $t = 3 \pm \epsilon$ and two roots at $t = 5 \pm \epsilon$, where $0 < \epsilon < 1$, ensuring that the polynomial has positive signs where needed. This is illustrated in Figure 4a.

In general, a row $\mathbf{y}_{i,:}^\pm$ with $\kappa_i$ sign changes requires exactly $\kappa_i$ roots, one in each interval between adjacent positions of opposite sign; the resulting polynomial has degree $\kappa_i \leq \kappa$. Padding its coefficient vector with zeros for the unused higher-order terms expresses every row within the common dimension $\kappa + 1$.

**Reconstructing the Adjacency Matrix** Stacking the coefficient vectors $\mathbf{h}_i$ as rows into $\mathbf{H} \in \mathbb{R}^{|\mathcal{E}||\mathcal{R}|\times(\kappa+1)}$, the entire sign matrix can be written as $\mathbf{Y}^\pm = \operatorname{sign}(\mathbf{H}\mathbf{E}^\top)$, which yields the desired decomposition with inner dimension $\kappa + 1$ and proves the theorem.

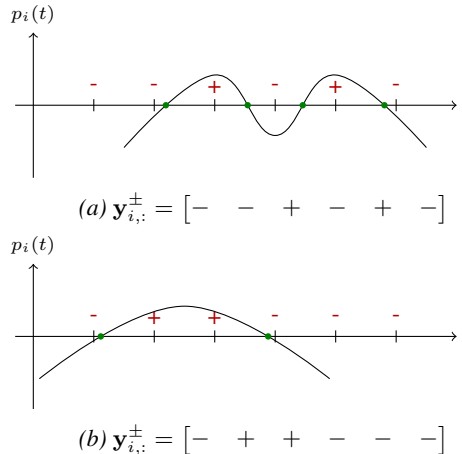

**Figure 4.** Fitting a polynomial $p_i(t)$ to the sign pattern of row $\mathbf{y}_{i,:}^{\pm}$. The sign of $p_i(t)$ at each integer $t = 1, \ldots, |\mathcal{E}|$ indicates the existence of an edge from $i$ to $t$. **(a) A pattern with $\kappa_i$ sign changes is represented by a polynomial of degree $\kappa_i$** (here $\kappa_i = 4$), whose $\kappa_i$ roots, shown as green dots, separate the sign regions. In the worst case the $c$ positive entries are isolated, giving $\kappa_i = 2c$. **(b) A better ordering of $\mathbf{y}_{i,:}^{\pm}$ groups the positive entries into blocks, lowering the number of sign changes** (here from 4 to 2) and hence the required degree.

**Bounding the number of sign changes** The dimension $\kappa + 1$ depends on the ordering $\boldsymbol{\pi}$, and the two corollaries follow by bounding $\kappa$ for specific orderings. First, a row with at most $c$ positive entries has at most $2c$ sign changes for *any* ordering – each isolated $+1$ contributes two changes – so $\kappa \leq 2c$ and $\mathrm{srank}(\mathbf{Y}^{\pm}) \leq 2c + 1$ (Corollary 4.2). Second, if an ordering groups the positive entries of each row into at most $b$ contiguous *blocks*, each block contributes at most two sign changes, so $\kappa \leq 2b$ and $\mathrm{srank}(\mathbf{Y}^{\pm}) \leq 2b + 1$ (Corollary **??**); since $b \leq c$, this is always at least as tight. Grouping is particularly effective for graphs with high clustering coefficients, where each cluster can be placed contiguously (Figure 4b), and for power-law degree distributions, where the ordering only needs to be optimised for the few high-degree entities. Finding the ordering that minimises $\kappa$ is, however, a hard problem that is dataset-specific.

**A heuristic ordering via Reverse Cuthill–McKee** We approximate this optimal ordering with the *Reverse Cuthill–McKee* (RCM) heuristic (Cuthill & McKee, 1969), which directly targets the block formulation above: making the $+1$ entries of each row as contiguous as possible (minimising the maximum number of blocks $b$) is an instance of matrix bandwidth minimisation, which is NP-hard but admits good and cheap heuristics. Concretely, we build a symmetric object–object co-occurrence matrix $\mathbf{C} \in \mathbb{N}^{|\mathcal{E}| \times |\mathcal{E}|}$, where the entry $c_{o,o'}$ counts the number of subject–relation pairs $(s, r)$ for which both $(s, r, o)$ and $(s, r, o')$ belong to the KG. Objects that often co-occur as answers to the same query thus receive a high weight and should be placed close

together in the ordering. In practice, we restrict the construction of $\mathbf{C}$ to the top $0.1\%$ most prolific $(s, r)$ rows (degree above the 99.9th percentile): in line with the power-law observation above, focusing the heuristic on the queries with the most answers yields tighter block counts than including the long tail of low-degree rows, which contribute few blocks regardless of $\boldsymbol{\pi}$ but dilute the co-occurrence signal. We then run RCM, a breadth-first reordering of the sparse graph induced by $\mathbf{C}$ that reduces its bandwidth, to obtain a single permutation $\boldsymbol{\pi}$ of the objects that clusters co-occurring objects together. Applying $\boldsymbol{\pi}$ to all rows, we count the resulting number of blocks per $(s, r)$ row and report their average and maximum in Table 2. The induced bound $2b+1$ is about $5\times$ tighter than the worst-case $2c+1$ on every dataset, giving a more practically meaningful estimate of the required dimension while remaining cheap to compute. As with the worst-case construction, this dimension corresponds to an interpolated, hand-constructed solution; models trained by gradient descent may need to overshoot it to converge (see the discussion at the end of Section 4).

**Interpreting the Embeddings** This decomposition suggests a natural interpretation in terms of node embeddings. Each object placed at position $t$ is the corresponding row of $\mathbf{E}$, i.e. the vector

$$\mathbf{e}_t = \begin{bmatrix} 1 & t & t^2 & \cdots & t^\kappa \end{bmatrix},$$

corresponding to evaluating the polynomial basis at $t$. Each subject–relation pair $i = (s, r)$ is the corresponding row $\mathbf{h}_i$ of $\mathbf{H}$, the coefficient vector defining the polynomial $p_i(t)$. The dot product $\mathbf{e}_t^\top \mathbf{h}_i = p_i(t)$ determines the sign at $(i, t)$, i.e., the existence of the triple $(s, r, o)$.

$\square$

# C. Subject prediction

Subject prediction is the task of predicting suitable subject entities for $(?, r, o)$ queries. There are conventionally two main approaches to subject prediction with LD-KGEs.

**Linear subject prediction** Bilinear models (see Section 2.2.1) can be used to score all subject entities against a relation-object pair as a vector-matrix multiplication:

$$\mathbf{z}_{:,r,o} = \mathbf{E}\mathbf{h}_{r,o} \tag{4}$$

where $\mathbf{h}_{r,o} \in \mathbb{R}^d$ is an embedded representation of the relation-object pair $(r, o)$ and $\mathbf{E} \in \mathbb{R}^{|\mathcal{E}| \times d}$ is the embedding matrix of all subject entity candidates. For example, in RESCAL's score function $\mathbf{e}_s^\top \mathbf{W}_r \mathbf{e}_o$ (Nickel et al., 2011), we have $\mathbf{h}_{r,o} = \mathbf{W}_r \mathbf{e}_o$, where $\mathbf{W}_r \in \mathbb{R}^{d \times d}$ is a matrix of shared parameters for relation $r$. In the score function $(\mathbf{e}_s \odot \mathbf{w}_r)^\top \mathbf{e}_o$ of DISTMULT (Yang et al., 2014), we have $\mathbf{h}_{r,o} = \mathbf{w}_r \odot \mathbf{e}_o$, where $\odot$ is the element-wise (Hadamard) product,

and $\mathbf{e}_s, \mathbf{w}_r, \mathbf{e}_o \in \mathbb{R}^d$. Under this perspective, our theoretical results on rank bottlenecks in LD-KGEs for solving object prediction (Section 3) can be extended to subject prediction by replacing the hidden state $\mathbf{h}_{s,r}$ with $\mathbf{h}_{r,o}$. However, our KGE-MoS solution is not applicable simultaneously for both object and subject prediction as the mixture of softmaxes breaks the bilinearity of the score function.

**Inverse relations** The most general approach which applies to all LD-KGEs– including neural network methods – is to introduce inverse relations $r^{-1}$ and to convert the subject prediction query $(?, r, o)$ into an object prediction query $(o, r^{-1}, ?)$ (Dettmers et al., 2018; Lacroix et al., 2018). This introduces $|\mathcal{R}|$ new relations and embeddings, but this parameter count is often negligible compared to the number of parameters of the entities.

$$\mathbf{z}_{:,r,o} := \mathbf{z}_{o,r^{-1},:} = \mathbf{h}_{o,r^{-1}}\mathbf{E}^\top \quad (5)$$

In this case, both the theoretical results and the KGE-MoS solution are immediately applicable. We can use the same KGE-MoS layer as follows:

$$P(S|o, r^{-1}) = \sum_{k=1}^{K} \pi_{o,r^{-1},k} \, \mathrm{softmax}(f_k(\mathbf{h}_{o,r^{-1}})\mathbf{E}^\top)$$
$$(6)$$

where $S$ is a random variable that ranges over the same entities as $O$.

# D. Background (cont.)

## D.1. Linear Object Prediction

In this section, we elaborate on how to rewrite the different LD-KGEs as a function linear in the object embedding $\phi(s, r, o) = \mathbf{h}_{s,r}^\top \mathbf{e}_o$, with $\mathbf{h}_{s,r}, \mathbf{e}_o \in \mathbb{R}^d$.

**Bilinear models** In RESCAL's score function $\mathbf{e}_s^\top \mathbf{W}_r \mathbf{e}_o$ (Nickel et al., 2011), we have $\mathbf{h}_{s,r}^\top = \mathbf{e}_s^\top \mathbf{W}_r$, where $\mathbf{W}_r \in \mathbb{R}^{d \times d}$ is a matrix of shared parameters for relation $r$. In the score function $(\mathbf{e}_s \odot \mathbf{w}_r)^\top \mathbf{e}_o$ of DISTMULT (Yang et al., 2014), we have $\mathbf{h}_{s,r}^\top = (\mathbf{e}_s \odot \mathbf{w}_r)^\top$, where $\odot$ is the element-wise (Hadamard) product, and $\mathbf{e}_s, \mathbf{w}_r, \mathbf{e}_o \in \mathbb{R}^d$. The same can be derived for COMPLEX (Trouillon et al., 2017), an extension of DISTMULT to complex-valued embeddings to handle asymmetry in the score function, or for CP (Lacroix et al., 2018) and SIMPLE (Kazemi & Poole, 2018), which use different embedding spaces for subjects and objects – though COMPLEX and SIMPLE with embedding sizes $d$ give forms $\mathbf{h}_{s,r}^\top \mathbf{e}_o$ with bottleneck dimension $2d$ as we detail in the next paragraph. Finally, TUCKER (Balazevic et al., 2019) decomposes the score tensor $\mathcal{Z} \in \mathbb{R}^{|\mathcal{E}| \times |\mathcal{R}| \times |\mathcal{E}|}$ as $\mathcal{Z} = \mathcal{W} \times_1 \mathbf{E} \times_2 \mathbf{R} \times_3 \mathbf{E}$ where $\mathcal{W} \in \mathbb{R}^{d \times d_r \times d}$ is a core tensor of shared parameters across all entities and relations

and $\times_n$ are mode-$n$ products.[4] $\mathbf{E} \in \mathbb{R}^{|\mathcal{E}| \times d}$, $\mathbf{R} \in \mathbb{R}^{|\mathcal{R}| \times d_r}$ are entity and relation embeddings, respectively. The score function is then $\phi_{\text{TUCKER}}(s, r, o) = \mathcal{W} \times_1 \mathbf{e}_s \times_2 \mathbf{w}_r \times_3 \mathbf{e}_o$. Let $\mathbf{h}_{s,r} = \mathcal{W} \times_1 \mathbf{e}_s \times_2 \mathbf{w}_r \in \mathbb{R}^d$. Then, $\phi_{\text{TUCKER}}(s, r, o) = (\mathbf{h}_{s,r}^{\text{TUCKER}})^\top \mathbf{e}_o$. Specific implementations of the core tensor $\mathcal{W}$ recovers the other bilinear models as special cases.

**COMPLEX, SIMPLE** As explained by (Kazemi & Poole, 2018; Balazevic et al., 2019), COMPLEX (Trouillon et al., 2017) can be seen as a special case of TUCKER and bilinear models by considering the real and imaginary part of the embedding concatenated in a single vector, e.g., $[\text{Re}(\mathbf{e}_o); \text{Im}(\mathbf{e}_o)] \in \mathbb{R}^{2d}$ for the object. We detail this to highlight that the rank bottleneck for these models is $2d$ rather than $d$. The score function of COMPLEX is

$$\begin{aligned}\phi_{\text{COMPLEX}}(s, r, o) = &(\text{Re}(\mathbf{e}_s) \odot \text{Re}(\mathbf{w}_r))^\top \text{Re}(\mathbf{e}_o) \\ &+ (\text{Im}(\mathbf{e}_s) \odot \text{Re}(\mathbf{w}_r))^\top \text{Im}(\mathbf{e}_o) \\ &+ (\text{Re}(\mathbf{e}_s) \odot \text{Im}(\mathbf{w}_r))^\top \text{Im}(\mathbf{e}_o) \\ &- (\text{Im}(\mathbf{e}_s) \odot \text{Im}(\mathbf{w}_r))^\top \text{Re}(\mathbf{e}_o)\end{aligned}$$

Let us define $\mathbf{h}_{sr}^1 = \text{Re}(\mathbf{e}_s) \odot \text{Re}(\mathbf{w}_r) - \text{Im}(\mathbf{e}_s) \odot \text{Im}(\mathbf{w}_r)$ and $\mathbf{h}_{sr}^2 = \text{Im}(\mathbf{e}_s) \odot \text{Re}(\mathbf{w}_r) + \text{Re}(\mathbf{e}_s) \odot \text{Im}(\mathbf{w}_r)$, with $\mathbf{h}_{sr}^1, \mathbf{h}_{sr}^2 \in \mathbb{R}^d$. We have

$$\phi_{\text{COMPLEX}}(s, r, o) = (\mathbf{h}_{sr}^1)^\top \text{Re}(\mathbf{e}_o) + (\mathbf{h}_{sr}^2)^\top \text{Im}(\mathbf{e}_o).$$

Concatenating the real and imaginary parts of $\mathbf{e}_o$ into a single vector $\mathbf{e}_o^{\text{COMPLEX}} = [\text{Re}(\mathbf{e}_o); \text{Im}(\mathbf{e}_o)] \in \mathbb{R}^{2d}$, and concatenating the hidden vectors $\mathbf{h}_{sr}^1$ and $\mathbf{h}_{sr}^2$ into a single vector $\mathbf{h}_{sr}^{\text{COMPLEX}} = [\mathbf{h}_{sr}^1; \mathbf{h}_{sr}^2] \in \mathbb{R}^{2d}$, we have

$$\phi_{\text{COMPLEX}}(s, r, o) = (\mathbf{h}_{sr}^{\text{COMPLEX}})^\top \mathbf{e}_o^{\text{COMPLEX}}$$

which is linear in the object embedding $\mathbf{e}_o^{\text{COMPLEX}}$, this time with bottleneck dimension $2d$. A similar trivial result can be done for SIMPLE (Kazemi & Poole, 2018), again considering the concatenation of two embeddings in a single vector in $\mathbb{R}^{2d}$.

**Neural KGEs** CONVE (Dettmers et al., 2018) scores triples as $f(\text{vec}(f([\mathbf{e}_s; \mathbf{w}_r] * \mathbf{w}))\mathbf{W})^\top \mathbf{e}_o$, with $f$ a non-linearity, $*$ a convolution operator, and vec a vectorization operator. The score function is linear in the object embedding $\mathbf{e}_o$. Similarly, STARE (Galkin et al., 2020) uses a score function Linear(SumPooling(Transformer($[\mathbf{e}_s + \mathbf{pe}[0]; \mathbf{w}_r + \mathbf{pe}[1]]$)))$^\top \mathbf{e}_o$, where $\mathbf{pe}$ are positional encodings, that has non-linearities in the transformer and pooling constructing $\mathbf{h}_{s,r}$ but that is still linear in the object embedding $\mathbf{e}_o$. R-GCNs (Schlichtkrull et al., 2018) and

---

[4]The mode-$n$ product $\mathcal{X} \times_n \mathbf{A}$ is the tensor obtained by multiplying each slice of $\mathcal{X}$ along the $n$-th mode by the corresponding column of $\mathbf{A}$. That is, $(\mathcal{X} \times_n \mathbf{A})_{i_1 \dots i_{n-1} j \, i_{n+1} \dots i_N} = \mathbf{a}_{j,:}^\top \mathbf{x}_{i_1, \dots, i_{n-1}, :, i_{n+1}, \dots, i_N}$.

COMPGCN (Vashishth et al., 2020) use powerful message passing schemes to build node representations for the entities $\mathbf{e}_s$ and $\mathbf{e}_o$, but still use a simple bilinear scoring function to score triples given the resulting representations – DISTMULT's score function giving the best performance. Language-based models like SIMKGC (Wang et al., 2022a) or MEMKGC (Choi et al., 2021) use BERT or other masked language models to encode powerful entity and relation embeddings $\mathbf{e}_s$, $\mathbf{e}_o$ and $\mathbf{w}_r$, but still use a linear projection to score objects in their final layer. COLE (Liu et al., 2022) uses two heads, one transformer and one BERT, and both use a linear output layer to score objects. LMKE (Wang et al., 2022b) is designed for triple classification but has a variant C-LMKE designed for link prediction, which uses a linear output layer where rows correspond to entities embeddings $\mathbf{e}_o$.

### D.2. Output functions in KGEs

In this section, we further detail the different output functions that LD-KGEs parameterised over the years and the loss functions that were used to train them.

**Raw scores with margin-based loss (RR)** Early KGEs (Bordes et al., 2013; Yang et al., 2014) used margin-based loss functions for ranking reconstruction (RR). The loss ensures that the scores of true triples are higher than the scores of false triples by at least a margin $\gamma > 0$, but does not ensure sign or calibrated reconstruction.

**Sigmoid layers with BCE loss (RR, SR, possibly DR)** (Trouillon et al., 2017), followed by (Dettmers et al., 2018), proposed to use binary cross-entropy (BCE) loss between the scores and the binary representation of the graph. They define a sigmoid layer $P(y_o = 1|\mathbf{h}_{s,r}) = \sigma(\mathbf{h}_{s,r}^{\top}\mathbf{e}_o)$, where $y_o = 1$ if $(s, r, o) \in \mathcal{G}$ and 0 otherwise. This defines a multi-label classifier with predictions $\hat{\mathbf{y}} = \mathbb{1}(\mathbf{Eh}_{s,r} > 0) \in \{0, 1\}^{|\mathcal{E}|}$ where $\mathbb{1}$ is the indicator function applied element-wise and the $i$-th entry is the binary prediction for object $i$. This classifier naturally aligns with sign-label reconstruction (SR). Calibrated scores (DR) were not a concern in this setting until (Arakelyan et al., 2021; 2023) started to combine the binary prediction scores of several simple queries to answer complex queries, which requires the scores to be calibrated.

**Softmax layers with CE loss (RR, possibly DR)** (Kadlec et al., 2017) proposed to use a cross-entropy (CE) loss. For a query $(s, r, ?)$, they define a vector of probabilities $\mathbf{p}_{s,r} = \operatorname{softmax}(\mathbf{Eh}_{s,r}) \in \Delta^{|\mathcal{E}|}$, with $\Delta^{|\mathcal{E}|}$ the probability simplex over the objects. Here, softmax is applied element-wise to get the components $p_{s,r,o} = \exp(\mathbf{h}_{s,r}^{\top}\mathbf{e}_o)/\sum_{o' \in \mathcal{E}} \exp(\mathbf{h}_{s,r}^{\top}\mathbf{e}_{o'})$. This vector should align with the data distribution (labels of the corresponding triples, normalised to sum to 1). Although

softmax is typically used for predicting a single correct class, here it is repurposed for multi-label scenarios by retrieving top-$k$ entries and ranking likely completions (RR). This modeling also opens the door to sampling (Loconte et al., 2023) which requires calibrated scores (DR).

In comprehensive benchmarking efforts, (Ruffinelli et al., 2020) and (Ali et al., 2022) re-evaluated all score functions with all possible output functions and loss functions for ranking reconstruction (RR). They found that softmax-based modeling generally outperform other approaches, closely followed by modeling with BCE loss. Models trained with margin-based loss functions were consistently the worst performing. Note that the authors did not evaluate the performance of models for sign-label reconstruction (SR) or calibrated score reconstruction (DR).

## E. Rank bottlenecks in KGEs (cont.)

### E.1. Log-linearity of the softmax bottleneck

As we mention in 3.3, the softmax function maps the $d$-dimensional linear subspace of scores to a smooth $d$-dimensional manifold within the $(|\mathcal{E}|-1)$-dimensional probability simplex $\Delta^{|\mathcal{E}|}$. While this manifold is non-linear, its shape is relatively simple and can be described by the following constraint: its composition with the log function is a linear function of the scores. An algebraic perspective on this bottleneck is provided by (Yang et al., 2018). Let $\mathbf{A} \in \mathbb{R}^{|\mathcal{E}||\mathcal{R}| \times |\mathcal{E}|}$ be the log-probability matrix where $a_{i,j} = \log P(O = j|\mathbf{h}_i)$. We have

$$\mathbf{A} = \begin{bmatrix} \log \operatorname{softmax}(\mathbf{Eh}_{1,1}) \\ \vdots \\ \log \operatorname{softmax}(\mathbf{Eh}_{|\mathcal{E}|,|\mathcal{R}|}) \end{bmatrix} \tag{7}$$

$$= \mathbf{HE}^{\top} - \begin{bmatrix} c_{1,1} \\ \vdots \\ c_{|\mathcal{E}|,|\mathcal{R}|} \end{bmatrix} \begin{bmatrix} 1 & \cdots & 1 \end{bmatrix} \tag{8}$$

where $c_i = \log \sum_{j=1}^{|\mathcal{E}|} \exp(\mathbf{Eh}_i)_j \in \mathbb{R}$ is the log-partition function. Since $\mathbf{HE}^{\top}$ has a rank at most $d$, and the term involving log-partition functions has a rank at most 1, the resulting log-probability matrix $\mathbf{A}$ has a rank of at most $d + 1$.

We can visualise this log-linearity using log-ratio transformations. For example, for a vector of probabilities $\mathbf{p} \in \Delta^{|\mathcal{E}|}$, we use an Additive Log-Ratio (ALR) transformation with $p_{|\mathcal{E}|}$ as the reference:

$$\operatorname{ALR}(\mathbf{p}) = \begin{bmatrix} \log \frac{p_1}{p_{|\mathcal{E}|}} \\ \vdots \\ \log \frac{p_{|\mathcal{E}|-1}}{p_{|\mathcal{E}|}} \end{bmatrix} \tag{9}$$

This transformation maps the simplex to a $d-1$ dimensional space where the log-partitions are eliminated in the ratios. As we visualise in Figure 5, the log-ratio of the probabilities span a linear subspace of the possible probability ratios. Any groundtruth that does not respect this log-linear constraint cannot be represented by the model.

### E.2. Ranking realisability bounds for bilinear KGEs from Wang et al. (2018)

As explained in Section 3.1, Theorem 3.2 gives a condition

$$d_{\text{RR}} \geq \text{srank}(\mathbf{Y}^{\pm}) - 1$$

for realising the exact rankings of a graph $\mathbf{Y}$ that lower bounds any KGE-specific condition. Wang et al. (2018) provide sufficient bounds for some specific bilinear KGEs. The tightest are for RESCAL and COMPLEX, which have the sufficient bound

$$d_{\text{RESCAL}}^{+} = d_{\text{COMPLEX}}^{+} = 2 \sum_{r \in \mathcal{R}} \text{rrank}(\mathbf{Y}_r)$$

where $\mathbf{Y}_r \in \{0,1\}^{|\mathcal{E}| \times |\mathcal{E}|}$ is the adjacency matrix for relation $r$. $\text{rrank}(\mathbf{Y}_r)$ is the rounding rank of $\mathbf{Y}_r$ with threshold $0.5$, which is the minimum rank of any real matrix $\mathbf{M}$ with $\text{round}(\mathbf{M}) = \mathbf{Y}_r$, where $\text{round}(x) = \mathbb{1}[x > 0.5]$ is applied element-wise and $\mathbb{1}[\cdot]$ is the indicator function.

Notice that the bound is a sufficiency bound, not a necessary one. The necessary and sufficient bound for RESCAL and COMPLEX are somewhere in between $d_{\text{RR}}$ and $d_{\text{RESCAL}}^{+}$ and are still unknown. Next, we confirm that this bound is (up to $2|\mathcal{R}|$ times) larger than the one in Theorem 3.2.

**Relating sign ranks and rounding ranks**  Neumann et al. (2016) show that the rounding rank of any boolean matrix $\mathbf{A} \in \{0,1\}^{n \times m}$ and sign rank of the corresponding sign matrix $\mathbf{A}^{\pm} = 2\mathbf{A} - 1 \in \{-1, +1\}^{n \times m}$ are tightly related. In particular, with the threshold $0.5$ it can be shown that $\text{rrank}(\mathbf{A})$ and $\text{srank}(\mathbf{A}^{\pm})$ differ by at most 1:

1. Any matrix $\mathbf{B}$ with $\text{round}(\mathbf{B}) = \mathbf{A}$ defines a sign matrix $\mathbf{C} := \mathbf{B} - \frac{1}{2}\mathbf{J}$ (where $\mathbf{J}$ is the all-ones matrix) that satisfies $\text{sign}(\mathbf{C}) = \mathbf{A}^{\pm}$. Therefore, if the rank of $\mathbf{B}$ is $r$, the the rank of $\mathbf{C}$ is at most $r+1$ (sum of $B$ and a rank-1 matrix). So $\text{srank}(\mathbf{A}^{\pm}) \leq \text{rrank}(\mathbf{A}) + 1$.

2. Conversely, any matrix $\mathbf{C}$ with $\text{sign}(\mathbf{C}) = \mathbf{A}^{\pm}$ defines a matrix $\mathbf{B} := \frac{1}{2}\mathbf{J} + \epsilon\mathbf{C}$ with $\epsilon > 0$ that satisfies $\text{round}(\mathbf{B}) = \mathbf{A}$. Therefore, if the rank of $\mathbf{C}$ is $r$, the the rank of $\mathbf{B}$ is at most $r+1$ (sum of a rank-1 matrix and $\mathbf{C}$). So $\text{rrank}(\mathbf{A}) \leq \text{srank}(\mathbf{A}^{\pm}) + 1$.

Therefore, $|\text{rrank}(\mathbf{A}) - \text{srank}(\mathbf{A}^{\pm})| \leq 1$.

**Expressing $d_{\text{RESCAL}}^{+}$ and $d_{\text{COMPLEX}}^{+}$ in terms of the sign rank**  Based on the above relationship between rounding ranks and sign ranks, we can express the bound from Wang et al. (2018) as

$$d_{\text{RESCAL}}^{+} = d_{\text{COMPLEX}}^{+} = 2 \sum_{r \in \mathcal{R}} \left( \text{srank}(\mathbf{Y}_r^{\pm}) - 1 \right)$$

where $\mathbf{Y}_r^{\pm} = 2\mathbf{Y}_r - 1 \in \{-1, +1\}^{|\mathcal{E}| \times |\mathcal{E}|}$ is the sign matrix for relation $r$.

**Relationship with $d_{\text{RR}}$**  Next, we relate $\sum_{r \in \mathcal{R}} \text{srank}(\mathbf{Y}_r^{\pm})$ to $\text{srank}(\mathbf{Y}^{\pm})$. $\mathbf{Y}^{\pm}$ can be written as the block matrix stacking each relation matrix $\mathbf{Y}^{\pm} = \begin{bmatrix} \mathbf{Y}_1^{\pm} \\ \vdots \\ \mathbf{Y}_{|\mathcal{R}|}^{\pm} \end{bmatrix}$. Then, $\text{srank}(\mathbf{Y}^{\pm}) = \sum_{r \in \mathcal{R}} \text{srank}(\mathbf{Y}_r^{\pm})$ only when all rows in all relation matrices are linearly independent. However, this might not be the case in practice. In the degenerate case where all the relation matrices are identical, we have $\text{srank}(\mathbf{Y}^{\pm}) = \text{srank}(\mathbf{Y}_1^{\pm})$, whereas $\sum_{r \in \mathcal{R}} \text{srank}(\mathbf{Y}_r^{\pm}) = |\mathcal{R}| \cdot \text{srank}(\mathbf{Y}_1^{\pm})$. In that case, the lower bound $d_{\text{RR}} \geq \text{srank}(\mathbf{Y}_1^{\pm}) - 1$ can be roughly $2|\mathcal{R}|$ times smaller than $d_{\text{RESCAL}}^{+} = d_{\text{COMPLEX}}^{+} = 2|\mathcal{R}|(\text{srank}(\mathbf{Y}_1^{\pm}) - 1)$.

## F. Experiments

### F.1. Datasets

We use the usual splits for FB15K237 (Toutanova & Chen, 2015), WN18RR (Dettmers et al., 2018) and ogbl-biokg (Hu et al., 2020). openbiolink (Breit et al., 2020) comes with four available datasets. We use the high-quality, directed set downloadable from the PyKeen library(Ali et al., 2022), which filters out test entities that do not appear in the training set and are not learnable by knowledge graph embedding models. Hetionet (Himmelstein et al., 2017) does not come with pre-defined splits. We obtain the splits via the PyKeen library using a seed of 42. Table 6 reports statistics for all datasets.

### F.2. Metrics

**Filtered NLL**  The negative log-likelihood (NLL) metric measures how well a model assigns probability to the true triples in the test set. However, in KGC, because the training and test sets are disjoint, a model trained to assign high probability to training triples might leave little probability mass for test triples, even if they follow similar patterns. In this case, the NLL can unfairly penalise models that fit the training set well but are able to generalise out-of-distribution, as generalisation in KGC is always *out-of-distribution* rather than in-distribution.

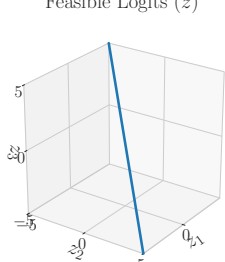
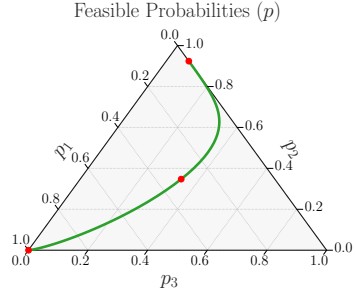
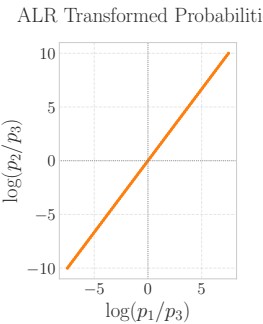

*Figure 5.* Visualisation of the softmax bottleneck. The blue line is the linear subspace of feasible scores for a fixed matrix $\mathbf{E}$, where each point is calculated for a different hidden state $\mathbf{h}_{s,r}$. The green line is the set of feasible probability distributions after the softmax function. The orange line is the additive-log ratio of the probabilities. **The feasible distributions are constrained to a manifold that is linear in the ALR space.** Any groundtruth that does not respect this constraint cannot be represented by the model.

*Table 6.* Dataset statistics and $(s, r, \cdot)$ out-degrees. The sufficient bound $d_{\text{SR}}^{+}$ for exact sign reconstruction is calculated based on the maximum out-degree according to Corollary 4.2.

| Dataset | #Entities | #Rels | #Triples | Inverse Relations | Out-Degree Mean | Out-Degree Median | Out-Degree Max | $d_{\text{SR}}^{+}$ |
|---|---|---|---|---|---|---|---|---|
| FB15k-237 | 14,541 | 237 | 310,116 | ✗ | 3.03 | 1 | 954 | 1,909 |
| | | | | ✓ | 3.83 | 1 | 4,364 | 8,729 |
| WN18RR | 40,943 | 11 | 93,003 | ✗ | 1.40 | 1 | 473 | 947 |
| | | | | ✓ | 1.70 | 1 | 510 | 1,021 |
| Hetionet | 45,158 | 24 | 2,250,197 | ✗ | 29.07 | 7 | 15,036 | 30,073 |
| | | | | ✓ | 21.66 | 6 | 15,036 | 30,073 |
| ogbl-biokg | 93,773 | 51 | 5,088,434 | ✗ | 40.54 | 14 | 29,328 | 58,657 |
| | | | | ✓ | 37.48 | 12 | 29,328 | 58,657 |
| openbiolink | 180,992 | 28 | 4,559,267 | ✗ | 14.47 | 2 | 2,251 | 4,503 |
| | | | | ✓ | 18.12 | 2 | 18,420 | 36,841 |

To address this issue, we use a filtered version of NLL when evaluating on the test set. For a $(s, r, ?)$ query, we zero out the probabilities of ground truth objects for that query seen during training, renormalising the probabilities of the model predictions. Formally, given a model prediction $P(o|s, r)$, we define the filtered test probability as

$$P^{\text{filtered}}(o|s, r) = \frac{\mathbb{1}[(s, r, o) \notin \mathcal{G}_{\text{train}}] P(o|s, r)}{\sum_{o' \in \mathcal{E}} \mathbb{1}[(s, r, o') \notin \mathcal{G}_{\text{train}}] P(o'|s, r)} \quad (10)$$

This filtered approach provides a more meaningful evaluation of the model's ability to generalise to unseen triples, as it focuses on the model's ability to distinguish test triples from truly negative samples rather than from training triples that the model has already learned. It prevents situations where a model that accurately assigns high probability to training instances might be unfairly disadvantaged compared to a less informed model when evaluating predictions on the disjoint test set (Figure 6).

**Ranking metrics** The quality of a KGC model is commonly assessed by ranking the scores of test triples

$(s, r, o) \in \mathcal{G}_{\text{test}}$. For example, in the context of object prediction, the ranks are computed as

$$R_\theta(s, r, o) := 1 + \sum_{e \in \mathcal{E}'} \mathbb{1}[\Phi_\theta(s, r, e) > \Phi_\theta(s, r, o)]. \quad (11)$$

where $\mathbb{1}[\cdot]$ is the indicator function and the set $\mathcal{E}' := \{e \in \mathcal{E} \mid (s, r, e) \notin \mathcal{G}\}$ filters the rank by only including object entities that do not form KG triples. A pessimistic version of the rank is sometimes considered by taking a non-strict inequality The mean reciprocal rank (MRR) is the mean of the reciprocal ranks MRR $= \frac{1}{|\mathcal{G}_{\text{test}}|} \sum_{(s,r,o) \in \mathcal{G}_{\text{test}}} R_\theta(s, r, o)^{-1}$. The mean rank (MR) is the arithmetic mean of the ranks, but is notably sensitive to outliers. The Hits@k metric is the proportion of test triples that are ranked within the top $k$ positions.

Notice that in ogbl-biokg, each test object for $(s, r, ?)$ are not ranked against all entities, but only against a predefined set of 500 entities. The same holds for subject prediction $(?, r, o)$.

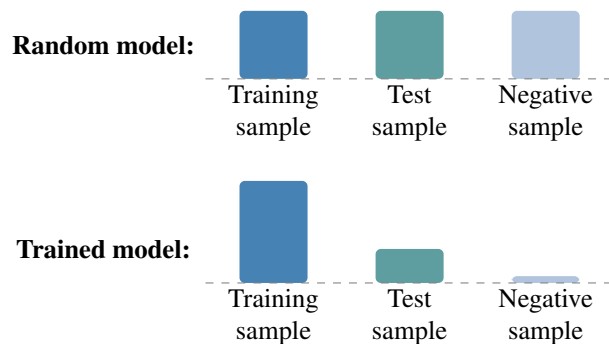

*Figure 6.* Likelihood assigned by models to training/test/random samples (conceptual graph). Because in KGC the set of training and test samples are disjoint, trained models leave less probability mass for test samples even if the model would assign higher probability to test samples if training samples were not options.

### F.3. Experimental setting

**General Hyperparameters** We guide the choice of hyperparameters by guidelines from the original papers of each model, as well as insights from recent benchmarking papers (Ruffinelli et al., 2020; Ali et al., 2022). All baseline models are trained using the Adam optimiser (Kingma & Ba, 2015) with a learning rate of $10^{-4}$, except for DISTMULT and COMPLEX (without KGE-MOS), which use a higher learning rate of $10^{-3}$ due to their shallow architecture. We use a batch size of 1000 for all models as KGEs are more stable with larger batch sizes (Ruffinelli et al., 2020), except for RESCAL, RESCAL-MOS, COMPGCN and COMPGCN-MOS, which use a batch size of 500 on `ogbl-biokg` and `openbiolink` for memory constraints. The models are trained for 30 training set epochs, with early stopping according to validation set metrics at a patience of 8 epochs.

**Regularisation** Following the findings of (Ruffinelli et al., 2020), which showed that dropout (Srivastava et al., 2014) is more effective than norm-based regularisation for KGEs, we apply a dropout of 0.1 to the encoded representation $\mathbf{h}_{s,r}$ (recalling that models can be represented as $\phi(s, r, o) = \mathbf{h}_{s,r}^{\top}\mathbf{e}_o$). An exception to this is CONVE, which already incorporates dropout within its neural network layers.

**CONVE specific hyperparameters** For CONVE, we adopt the hyperparameter settings from its original paper (Dettmers et al., 2018). The model reshapes the input subject and relation embeddings into two-dimensional "images" (height 8), followed by a two-layer convolutional network with 32 channels and a kernel size of $(3, 3)$. The output is then passed through a fully connected layer to obtain $\mathbf{h}_{s,r}$. Regularisation for CONVE includes dropout on embeddings (0.2), feature maps after convolution (0.3), and the output of the fully connected layer (0.3). Each layer in CONVE is also followed by a batch normalisation layer (Ioffe

& Szegedy, 2015).

**COMPGCN specific hyperparameters** For COMPGCN, we use the following hyperparameters: initial embedding dimension of 100 for entities and relations and GCN dimension (which dictates the bottleneck) of $d = 200$, $d = 100$ or $d = 50$ depending on the dataset. We use a single GCN layer without bias terms and use COMPGCN without basis decomposition. Dropout is applied at two stages: 0.1 dropout in the GCN layer and 0.1 hidden dropout after the GCN layer. For the composition operation in COMPGCN, we use DISTMULT as was empirically shown to be the best option in the original paper (Vashishth et al., 2020).

**KGE-MOS details and hyperparameters** For the KGE-MOS layer, defined as:

$$P(O|s, r) = \sum_{k=1}^{K} \pi_{s,r,k} \, \mathrm{softmax}(f_k(\mathbf{h}_{s,r})\mathbf{E}^{\top}) \quad (12)$$

with $\sum_{k=1}^{K} \pi_{s,r,k} = 1$, we use two-layer projections for $f_k(\mathbf{h}_{s,r})$ of dimension $d$ (the same as the KGE dimension). Each projection layer is followed by batch normalisation (Ioffe & Szegedy, 2015), a non-linear activation, and hidden dropout (applied in that order). To encourage the use of all mixture components, we incorporate the entropy of the distribution defined by the mixture weights $(\pi_{s,r,1}, \ldots, \pi_{s,r,K})$ into the loss function. Our hyperparameter search for KGE-MOS involved a random exploration over learning rate ($10^{-2}$ to $10^{-5}$, log-scale), hidden dropout (0.0 to 0.2), activation type (ReLU, LeakyReLU, GeLU, tanh), and entropy regularisation weight ($10^{-7}$ to 1.0, log-scale). The final reported results for KGE-MOS were obtained with a learning rate of $10^{-4}$, hidden dropout of 0.1, LeakyReLU activation, and an entropy regularisation weight of $10^{-3}$.

**Parameters initialisation** Parameters for the KGEs and all model layers are initialised using Xavier uniform initialisation (Glorot & Bengio, 2010). Repeated runs are performed with different initialisation seeds for statistical significance.

**Hardware** All experiments are run on NVIDIA A10 GPUs with 24GB of memory. Time of execution for each baseline ranges from a couple of hours for `FB15K237`, the smallest dataset, to 1-2 days for `openbiolink`, the largest dataset.

### F.4. Additional results

F.4.1. STANDARD DEVIATIONS AND HITS @ METRICS

Tables 11 to 15 report full results for `FB15k-237`, `Hetionet`, `ogbl-biokg`, `openbiolink`, and `WN18RR`.

*Table 7.* Empirical rank of the log-probability matrix for DISTMULT-MOS on the test set of `FB15k-237`.

| Model | Empirical rank |
|---|---|
| DISTMULT-MOS, $K = 1$, $d = 200$ | 201 |
| DISTMULT-MOS, $K = 4$, $d = 200$ | 3,651 |
| DISTMULT-MOS, $K = 1$, $d = 1000$ | 1,001 |
| DISTMULT-MOS, $K = 4$, $d = 1000$ | 14,541 |

Note that results for COMPLEX use halved embedding sizes $d = 100$ and $d = 500$ because it has real and imaginary parts leading to a rank bottleneck of $2d$ (see Appendix D.1).

On the other hand, while results for COMPGCN use $d = 200$ on `FB15k-237` and `Hetionet` and are comparable to the other models, they are conducted with $d = 100$ on `ogbl-biokg`, and $d = 50$ on `openbiolink` due to computational constraints.

#### F.4.2. EMPIRICAL RANK

Borrowing the idea of Yang et al. (2018), we confirm empirically that KGE-MOS breaks the rank bottleneck by measuring the rank of the log-probability matrix measured on the test data for $K = 1$ softmax and $K > 1$ softmaxes. That is, we compute the rank of

$$\begin{bmatrix} \log P(O|sr_1) \\ \vdots \\ \log P(O|sr_{N_{\text{test}}}) \end{bmatrix}$$

where $sr_i$ ranges over subject relation pairs in the test set.

We find the empirical ranks in Table 7. When $K = 1$, the rank is as expected $d + 1$ (where $d$ is the rank of the logits and 1 is the rank of the log partition matrix). **Increasing $K$ quickly increases the empirical rank.** In fact, as in `FB15k-237`, the number of entities $|\mathcal{E}| = 14,951$, the matrix is even **full rank** when $d = 1000$ and $K > 1$. In other words, the bottleneck is effectively broken.

#### F.4.3. INFERENCE TIME

In Table 8 we report inference time results for all methods.

We also report the total wall-clock time to complete 30 training epochs on `openbiolink`, our largest dataset, at $d = 1000$. Table 9 shows the two baselines with the largest (DISTMULT) and smallest (CONVE) relative overhead from KGE-MOS. The KGE-MOS variants are roughly $2\times$ slower to train, consistent with the $1.69$–$2.75\times$ per-minibatch slowdown reported above.

*Table 8.* Inference and backpropagation time for a single minibatch on `openbiolink` at $d = 1000$. RESCAL and RESCAL-MOS are compared at batch size 500 to fit on device memory. Other models are compared at batch size 1000.

| Model | Time per batch (ms) |
|---|---|
| DISTMULT | 1.34 |
| DISTMULT-MOS ($K = 1$) | 3.30 |
| DISTMULT-MOS ($K = 4$) | 3.69 |
| RESCAL | 1.51 |
| RESCAL-MOS ($K = 1$) | 3.49 |
| RESCAL-MOS ($K = 4$) | 3.42 |
| CONVE | 2.72 |
| CONVE-MOS ($K = 1$) | 4.74 |
| CONVE-MOS ($K = 4$) | 4.60 |

*Table 9.* Total wall-clock time for 30 training epochs on `openbiolink` at $d = 1000$, for the baselines with the largest (DISTMULT) and smallest (CONVE) relative overhead from KGE-MOS (using $K = 4$). The $\sim 2\times$ overhead matches the per-minibatch slowdown of Table 8.

| Model | Training time | vs. base |
|---|---|---|
| DISTMULT | 21h22m | — |
| DISTMULT-MOS ($K = 4$) | 43h03m | $2.02\times$ |
| CONVE | 25h25m | — |
| CONVE-MOS ($K = 4$) | 47h11m | $1.86\times$ |

#### F.4.4. BOTTLENECKS IN HIGHER-DIMENSIONAL SETTINGS ON `OGBL-BIOKG`

Next, we target a negative result: we evaluate whether KGE-MOS still shows a significant performance gain when the embedding dimension $d$ is sufficiently large. Indeed, for a sufficiently large dimension $d$ (dataset dependent), the bottleneck impact is not as great. Our results on `FB15K-237` and `WN18RR` (small datasets) aim to demonstrate this. We try to replicate this insight on one of the larger datasets (`ogbl-biokg`, the most densely connected one) with $d = 5000$.

We had to reduce our batch size from 1000 to 200 samples to accommodate for the larger dimension in the GPU memory (24GB). We measure two baselines, (i) DISTMULT-MOS with $K = 1$, (ii) DISTMULT-MOS with $K = 4$. DISTMULT-MOS with $K = 1$ does not break the rank bottleneck, but is included to isolate the effect of adding an asymmetric layer to DISTMULT vs breaking the rank bottleneck (see our paragraph "Mixture Ablation" in Section 6.1).

The results gives us two findings. (i) Breaking the rank bottleneck does not notably improve the performance on `ogbl-biokg` when $d = 5000$ (DISTMULT-MOS, $K = 4 \approx$ DISTMULT-MOS, $K = 1$). (ii) However, lowering the batch size to fit the model at $d = 5000$ in memory strongly deterio-

*Table 10.* Results on `ogbl-biokg` at $d = 5000$. Breaking rank bottlenecks does not improve performance when the dimension is sufficiently large. However, increasing $d$ comes at the cost of sacrificing critical hyperparameters such as batch size due to GPU memory constraints which deteriorates performance.

| **Model** ($d = 5000$) | **NLL** | **MRR** | **Param** |
|---|---|---|---|
| DISTMULT-MOS, $K = 1$ | 4.54 | 0.79 | 519.4M |
| DISTMULT-MOS, $K = 4$ | 4.54 | 0.79 | 669.5M |

rates the performance compared to the baselines at $d = 1000$ (compare these results with those of Table 13). In other words, **the simple solution of increasing the embedding dimension, in addition to using vastly more parameters, comes at the cost of sacrificing critical hyperparameters such as batch size which deteriorates performance**.

### F.4.5. EFFECT OF THE PROJECTION FUNCTION

We extend the ablation of Section 6.1 (Table 4) to also compare KGE-MOS using a single-layer projection $f_k$ against the two-layer projection used in our main experiments (Section 5). Table 17 reports NLL and MRR on `ogbl-biokg` at $d = 1000$ for DISTMULT-MOS and COMPLEX-MOS, the two best-performing models, across both the number of softmaxes $K$ and the projection depth. We observe two consistent trends: (i) a more expressive projection helps (the 2-layer projection outperforms the 1-layer one at every $K$), and (ii) increasing the number of softmaxes $K$ helps (larger $K$ outperforms smaller $K$). While the mixture is what enables the geometric non-linearity that breaks the rank bottleneck, accurate per-component projections also matter, as they add precision to the mixed distributions.

### F.4.6. TRANSLATION- AND ROTATION-BASED BASELINES

We expand on the discussion of Section 7 for translation- and rotation-based KGEs such as TransE (Bordes et al., 2013) and RotatE (Sun et al., 2019), which score objects with a Euclidean distance rather than a linear decoder.

**Expressivity**  Structured bounds for universality and realisability in RR/SR/DR remain to be explored for these models. The one result we are aware of concerns SR universality, for which the lower bound is at least $d \geq |\mathcal{E}| - 1$. This follows from the VC dimension of hypersphere classifiers (Dudley, 1979), whose decision regions are balls defined by a centre and a radius – in distance-based KGEs, each object embedding plays the role of a centre and the score threshold that of a radius. Shattering (i.e., correctly classifying) an arbitrary combination of $|\mathcal{E}|$ objects then requires $d \geq |\mathcal{E}| - 1$. This is only a sanity check for SR universality and is likely not sufficient, since universality requires shattering all arbitrary

object combinations simultaneously, not just one. Still, this dimension is already close to the $d \geq |\mathcal{E}|$ needed by linear-decoder LD-KGEs (Theorem 3.3). In other words, on this one known result, changing the score-function geometry to a Euclidean one does not improve expressivity. Together with new theoretical results, investigating a translational extension of KGE-MOS is an interesting direction for future work.

**Empirical comparison**  We compare LD-KGEs against TransE and RotatE on `ogbl-biokg` at $d = 1000$. As anticipated, RotatE performs comparably to the bottlenecked DISTMULT baseline, while TransE is slightly worse (Table 18). Both are clearly outperformed by DISTMULT-MOS, which breaks the rank bottleneck.

### F.4.7. NON-LINEAR CONCATENATION DECODERS

One way to escape the rank bottleneck is to replace the linear decoder of LD-KGEs with a non-linear one. We consider *ConcatMLP*, which scores a triple by applying a multi-layer perceptron (MLP) to the concatenation of the subject, relation, and object embeddings,

$$\phi(s, r, o) = \text{MLP}\big([\mathbf{e}_s; \mathbf{w}_r; \mathbf{e}_o]\big). \qquad (13)$$

Since $\phi$ is non-linear in $\mathbf{e}_o$, ConcatMLP is not constrained by the rank bottleneck and can in principle represent arbitrary score functions. Its limitation is computational and specific to object prediction.

**Efficient scoring**  Whereas LD-KGEs score all $|\mathcal{E}|$ candidate objects of a query with a single vector-matrix product $\mathbf{h}_{s,r}\mathbf{E}^\top$, a non-linear decoder must in principle evaluate the MLP separately for every candidate object. The first layer is linear, however, so the object-dependent term can be shared. Writing the first weight matrix in blocks as $W_1 = [W_1^s \mid W_1^r \mid W_1^o]$, we precompute the object projection $M = \mathbf{E}(W_1^o)^\top \in \mathbb{R}^{|\mathcal{E}| \times H}$ once for all objects (as for the embedding table), and per query the term $\mathbf{a}_{s,r} = W_1^s \mathbf{e}_s + W_1^r \mathbf{w}_r + \mathbf{b}_1$. All object scores are then obtained by applying the activation and the remaining layers to $M + \mathbf{a}_{s,r}$, batched over objects. This brings the cost to $O(B\,|\mathcal{E}|)$ for a batch of $B$ queries – the same complexity class as LD-KGEs– with a per-(query, object) cost that is roughly constant across graph sizes (about $0.04\,\mu$s for graphs from 14.5k to 94k entities).

**Limitations**  Despite this, two obstacles make ConcatMLP impractical for the large output spaces we target. First, the constant factor is large and grows with the MLP width (Table 19): on `FB15k-237` at $d = 200$, a forward pass is between $50\times$ and $931\times$ slower than DISTMULT. Second, and more importantly, scoring all objects materialises a $(B, |\mathcal{E}|, H)$ hidden tensor – a factor $H$ larger than the

*Table 11.* Full results for the FB15k-237 dataset.

| Model | FB15k-237 | | | | | | |
|---|---|---|---|---|---|---|---|
| | NLL↓ | MRR↑ | MR↓ | Hits@1↑ | Hits@3↑ | Hits@10↑ | Param |
| *d* = 200 | | | | | | | |
| DISTMULT | 4.74±.08 | .304±.004 | 228±7.5 | .216±.003 | .331±.005 | .482±.004 | 3.0M |
| DISTMULT-MOS | 4.65±.01 | .306±.002 | 214±0.3 | .220±.002 | .336±.001 | .479±.003 | 3.3M |
| COMPLEX † | 4.74±.01 | .303±.001 | 220±2.3 | .216±.001 | .330±.002 | .482±.001 | 3.0M |
| COMPLEX-MOS † | 4.71±.01 | .301±.001 | 221±1.4 | .218±.001 | .329±.002 | .467±.001 | 3.3M |
| RESCAL | 4.79±.01 | .285±.001 | 246±2.3 | .210±.001 | .309±.001 | .432±.001 | 21.9M |
| RESCAL-MOS | 4.65±.01 | .318±.001 | 220±1.1 | .230±.002 | .348±.001 | .494±.001 | 22.2M |
| CONVE | **4.48±.01** | **.321±.001** | **176±2.6** | **.232±.000** | **.351±.001** | **.499±.001** | 5.1M |
| CONVE-MOS | 4.57±.01 | .311±.002 | 203±1.1 | .227±.002 | .339±.003 | .479±.002 | 5.4M |
| COMPGCN | 4.80±.07 | .300±.005 | 250±19.8 | .215±.004 | .328±.005 | .474±.008 | 1.6M |
| COMPGCN-MOS | 4.86±.03 | .310±.001 | 223±5.8 | .222±.001 | .339±.001 | .486±.001 | 2.0M |
| *d* = 1000 | | | | | | | |
| DISTMULT | **4.56±.01** | **.331±.001** | 208±2.8 | **.241±.001** | **.362±.001** | **.514±.003** | 15.0M |
| DISTMULT-MOS | 4.72±.00 | .311±.001 | 231±2.3 | .221±.003 | .341±.000 | .492±.002 | 23.0M |
| COMPLEX † | 4.71±.45 | .317±.027 | 224±72.7 | .231±.019 | .347±.031 | .491±.047 | 15.0M |
| COMPLEX-MOS † | 4.64±.00 | .314±.001 | 226±3.1 | .225±.001 | .343±.001 | .497±.001 | 23.0M |
| RESCAL | 4.64±.00 | .307±.001 | 216±0.7 | .221±.001 | .335±.003 | .483±.002 | 488.5M |
| RESCAL-MOS | 4.63±.01 | .325±.002 | 259±5.4 | .236±.004 | .357±.002 | .505±.001 | 496.5M |
| CONVE | 4.65±.04 | .301±.001 | **185±4.3** | .212±.001 | .329±.001 | .485±.002 | 70.1M |
| CONVE-MOS | 4.69±.01 | .316±.001 | 222±1.5 | .227±.001 | .347±.001 | .497±.002 | 78.2M |

† Results for COMPLEX use halved embedding sizes *d* = 100 and *d* = 500.

*Table 12.* Full results for the Hetionet dataset.

| Model | Hetionet | | | | | | |
|---|---|---|---|---|---|---|---|
| | NLL↓ | MRR↑ | MR↓ | Hits@1↑ | Hits@3↑ | Hits@10↑ | Param |
| *d* = 200 | | | | | | | |
| DISTMULT | 6.10±.00 | .250±.001 | 695±2.4 | .176±.001 | .274±.001 | .395±.001 | 9.0M |
| DISTMULT-MOS | **5.83±.00** | **.277±.002** | **488±1.3** | **.202±.003** | **.303±.002** | **.423±.002** | 9.4M |
| COMPLEX † | 6.10±.00 | .249±.001 | 698±2.8 | .174±.001 | .274±.001 | .395±.000 | 9.0M |
| COMPLEX-MOS † | 5.85±.00 | .269±.001 | 500±2.9 | .194±.001 | .294±.001 | .415±.001 | 9.4M |
| RESCAL | 6.13±.00 | .219±.001 | 607±0.4 | .153±.001 | .237±.001 | .351±.001 | 10.9M |
| RESCAL-MOS | 5.87±.01 | .274±.000 | 505±2.0 | .200±.000 | .300±.001 | .419±.001 | 11.3M |
| CONVE | 6.03±.00 | .252±.001 | 610±2.3 | .180±.001 | .275±.002 | .395±.002 | 11.1M |
| CONVE-MOS | 5.92±.00 | .263±.001 | 536±1.1 | .193±.002 | .284±.002 | .400±.001 | 11.4M |
| COMPGCN | 5.95±.01 | .260±.005 | 542±3.4 | .185±.005 | .285±.006 | .409±.006 | 4.7M |
| COMPGCN-MOS | 5.86±.08 | .266±.012 | 484±16.0 | .191±.010 | .292±.015 | .414±.018 | 5.0M |
| *d* = 1000 | | | | | | | |
| DISTMULT | 6.04±.00 | .288±.000 | 633±1.9 | .217±.001 | .314±.000 | .429±.000 | 45.2M |
| DISTMULT-MOS | 5.76±.00 | .312±.001 | **491±1.5** | .233±.001 | **.344±.001** | **.467±.001** | 53.2M |
| COMPLEX † | 5.99±.00 | .292±.000 | 658±1.8 | .219±.001 | .320±.000 | .437±.000 | 45.2M |
| COMPLEX-MOS † | 5.78±.00 | .303±.000 | 504±2.1 | .223±.001 | .335±.001 | .459±.001 | 53.2M |
| RESCAL | 5.93±.00 | .243±.001 | 650±3.6 | .165±.001 | .270±.001 | .398±.001 | 93.1M |
| RESCAL-MOS | 5.87±.03 | .300±.009 | 542±4.9 | .223±.009 | .327±.010 | .454±.008 | 101.2M |
| CONVE | 6.06±.00 | .262±.001 | 635±4.7 | .187±.001 | .288±.001 | .411±.001 | 100.3M |
| CONVE-MOS | **5.71±.01** | **.313±.001** | 505±3.7 | **.237±.002** | .343±.001 | .462±.001 | 108.3M |

† Results for COMPLEX use halved embedding sizes *d* = 100 and *d* = 500.

$(B, |\mathcal{E}|)$ scores of LD-KGEs– which must be retained for backpropagation. This memory blow-up is the binding constraint: on `ogbl-biokg` ConcatMLP runs out of memory already at $B = 64$, and on `FB15k-237` the largest architecture ($[1024, 512]$) runs out of memory on the backward pass at $B = 128$. Training is then only possible by sub-sampling a few objects per step, which is at odds with the dense, large-scale object prediction we study. Con-

cretely, on `FB15k-237` ($d = 200$, $B = 128$) a minibatch takes DISTMULT $0.20$ ms (forward) and $0.66$ ms (forward and backward), versus $74.0$ ms and $182.3$ ms for a $[512, 256]$ ConcatMLP – a $364\times$ and $278\times$ overhead, respectively.

This is why non-linear concatenation decoders – and, for related reasons, fully conditional models such as NBFNet (Zhu et al., 2021) – are typically reserved for settings with a small output space, such as triple classification

*Table 13.* Full results for the ogbl-biokg dataset.

| Model | ogbl-biokg | | | | | | |
|---|---|---|---|---|---|---|---|
| | NLL↓ | MRR↑ | MR↓ | Hits@1↑ | Hits@3↑ | Hits@10↑ | Param |
| $d = 200$ | | | | | | | |
| DISTMULT | 4.83±.01 | .792±.001 | 5.60±0.1 | .713±.001 | .849±.001 | **.935±.001** | 18.8M |
| DISTMULT-MOS | 4.65±.00 | .792±.001 | 5.32±0.0 | .716±.002 | .844±.000 | .930±.000 | 19.1M |
| COMPLEX [†] | 4.83±.01 | .792±.001 | 5.56±0.0 | .713±.002 | .849±.001 | .935±.000 | 18.8M |
| COMPLEX-MOS [†] | **4.65±.00** | **.793±.001** | **5.26±0.0** | **.717±.002** | **.845±.001** | .931±.000 | 19.1M |
| RESCAL | 4.89±.00 | .763±.001 | 6.01±0.0 | .679±.001 | .818±.001 | .917±.000 | 22.8M |
| RESCAL-MOS | 4.70±.00 | .780±.001 | 5.47±0.0 | .699±.001 | .835±.001 | .928±.000 | 23.2M |
| CONVE | 4.94±.00 | .782±.001 | 5.57±0.0 | .701±.002 | .838±.001 | .928±.000 | 20.8M |
| CONVE-MOS | 4.77±.01 | .768±.002 | 5.77±0.1 | .683±.003 | .824±.002 | .923±.001 | 21.2M |
| COMPGCN [‡] | 5.11±.20 | .749±.019 | 7.00±0.7 | .665±.021 | .801±.019 | .905±.012 | 9.5M |
| COMPGCN-MOS [‡] | 4.96±.15 | .744±.022 | 6.93±0.9 | .659±.025 | .799±.022 | .904±.014 | 9.6M |
| $d = 1000$ | | | | | | | |
| DISTMULT | 4.89±.02 | .801±.002 | 5.96±0.1 | .726±.003 | .855±.002 | .936±.001 | 93.9M |
| DISTMULT-MOS | **4.34±.00** | **.837±.001** | **4.51±0.0** | **.772±.002** | **.884±.000** | **.951±.000** | 101.9M |
| COMPLEX [†] | 4.86±.00 | .806±.001 | 5.49±0.0 | .731±.002 | .860±.001 | .940±.000 | 93.9M |
| COMPLEX-MOS [†] | 4.39±.00 | .836±.001 | 4.50±0.0 | .770±.001 | .883±.000 | .951±.000 | 101.9M |
| RESCAL | 4.74±.00 | .799±.001 | 5.66±0.0 | .723±.001 | .851±.000 | .939±.000 | 195.8M |
| RESCAL-MOS | 4.42±.01 | .824±.004 | 4.87±0.1 | .755±.005 | .874±.002 | .947±.001 | 203.8M |
| CONVE | 4.93±.00 | .807±.000 | 5.25±0.0 | .731±.001 | .863±.000 | .941±.000 | 149.0M |
| CONVE-MOS | 4.43±.00 | .817±.001 | 5.17±0.1 | .744±.001 | .872±.000 | .946±.000 | 157.0M |

[†] Results for COMPLEX use halved embedding sizes $d = 100$ and $d = 500$.
[‡] Results for COMPGCN use $d = 100$ due to computational constraints.

*Table 14.* Full results for the openbiolink dataset.

| Model | openbiolink | | | | | | |
|---|---|---|---|---|---|---|---|
| | NLL↓ | MRR↑ | MR↓ | Hits@1↑ | Hits@3↑ | Hits@10↑ | Param |
| $d = 200$ | | | | | | | |
| DISTMULT | 5.14±.00 | .302±.002 | 1120±10.6 | .195±.004 | .342±.000 | .530±.000 | 36.2M |
| DISTMULT-MOS | **5.03±.01** | .314±.001 | 966±33.4 | .207±.001 | .351±.001 | .543±.000 | 36.5M |
| COMPLEX [†] | 5.13±.01 | .301±.001 | 1200±20.2 | .193±.002 | .340±.001 | .530±.001 | 36.2M |
| COMPLEX-MOS [†] | 5.06±.00 | .313±.002 | 909±18.5 | .206±.003 | .350±.002 | .539±.004 | 36.5M |
| RESCAL | 5.16±.00 | .303±.001 | 960±2.20 | .197±.002 | .339±.001 | .527±.001 | 38.4M |
| RESCAL-MOS | 5.04±.01 | **.323±.005** | 949±78.8 | **.217±.004** | **.361±.005** | **.547±.005** | 38.8M |
| CONVE | 5.28±.02 | .286±.000 | 794±17.6 | .181±.001 | .320±.002 | .509±.002 | 38.3M |
| CONVE-MOS | 5.10±.01 | .304±.002 | 967±47.9 | .200±.002 | .340±.003 | .528±.002 | 38.6M |
| COMPGCN [‡] | 5.43±.07 | .263±.007 | 573±27.0 | .162±.005 | .293±.009 | .480±.012 | 18.3M |
| COMPGCN-MOS [‡] | 5.35±.06 | .278±.001 | **515±35.9** | .174±.003 | .312±.004 | .495±.009 | 18.3M |
| $d = 1000$ | | | | | | | |
| DISTMULT | 5.17±.02 | .316±.004 | 1210±21.2 | .209±.004 | .357±.003 | .541±.004 | 181.0M |
| DISTMULT-MOS | 4.89±.01 | **.347±.000** | 799±32.1 | **.236±.002** | **.392±.002** | **.579±.002** | 189.0M |
| COMPLEX [†] | 5.12±.01 | .322±.001 | 1230±33.9 | .213±.001 | .364±.002 | .550±.001 | 181.0M |
| COMPLEX-MOS [†] | **4.87±.01** | .345±.001 | **725±18.7** | .235±.002 | .388±.001 | .575±.001 | 189.0M |
| RESCAL | 5.03±.00 | .328±.002 | 1330±26.6 | .222±.002 | .366±.001 | .552±.002 | 237.0M |
| RESCAL-MOS | 5.00±.01 | .330±.003 | 1210±27.8 | .220±.003 | .373±.003 | .559±.004 | 245.0M |
| CONVE | 5.24±.00 | .308±.001 | 1140±24.2 | .201±.001 | .348±.001 | .535±.001 | 236.2M |
| CONVE-MOS | 4.91±.01 | .336±.000 | 881±22.4 | .227±.001 | .378±.000 | .565±.001 | 244.2M |

[†] Results for COMPLEX use halved embedding sizes $d = 100$ and $d = 500$.
[‡] Results for COMPGCN use $d = 50$.

or relation prediction. KGE-MOS instead breaks the rank bottleneck while preserving the efficient per-component vector-matrix products of LD-KGEs, with activation memory growing only as $O(K |\mathcal{E}|)$ for $K \ll H$ softmax components.

### F.5. Sign reconstruction

Our experiments (Section 6.1) evaluate models on ranking reconstruction (RR), through the MRR, and distributional reconstruction (DR), through the filtered NLL. We did not include sign reconstruction (SR) in the main evaluation for the following reason. SR is a binary classification task over

*Table 15.* Full results for the WN18RR dataset.

| Model | WN18RR | | | | | | |
|---|---|---|---|---|---|---|---|
| | NLL↓ | MRR↑ | MR↓ | Hits@1↑ | Hits@3↑ | Hits@10↑ | Param |
| *d = 200* | | | | | | | |
| DISTMULT | 6.73±.02 | .421±.001 | 7220±2.5 | .397±.002 | .429±.001 | .469±.001 | 8.2M |
| DISTMULT-MOS | 9.08±.79 | .129±.154 | 8080±2000 | .107±.152 | .134±.159 | .171±.157 | 8.6M |
| COMPLEX † | **6.04±.04** | **.436±.001** | 7640±347 | **.413±.002** | **.446±.001** | **.479±.001** | 8.2M |
| COMPLEX-MOS † | 8.67±.38 | .143±.067 | 7590±563 | .112±.062 | .153±.071 | .202±.074 | 8.6M |
| RESCAL | 8.66±.54 | .215±.113 | 7640±2010 | .187±.116 | .225±.112 | .267±.104 | 9.1M |
| RESCAL-MOS | 8.91±.56 | .139±.137 | 7630±1730 | .112±.130 | .146±.145 | .188±.148 | 9.4M |
| CONVE | 7.31±.03 | .230±.002 | **4300±265** | .177±.003 | .250±.002 | .327±.004 | 10.3M |
| CONVE-MOS | 8.54±.41 | .116±.071 | 5610±1390 | .078±.056 | .127±.081 | .189±.102 | 10.6M |
| COMPGCN | 7.38±.13 | .411±.002 | 4550±167 | .386±.002 | .418±.002 | .459±.002 | 4.2M |
| COMPGCN-MOS | 9.04±.04 | .396±.001 | 5250±120 | .379±.002 | .402±.001 | .428±.003 | 4.6M |
| *d = 1000* | | | | | | | |
| DISTMULT | 6.44±.01 | .438±.001 | 6380±70.9 | .405±.001 | .449±.002 | .507±.002 | 41.1M |
| DISTMULT-MOS | 6.02±.02 | .435±.002 | 5650±106 | .413±.002 | .443±.002 | .477±.002 | 49.2M |
| COMPLEX † | 5.67±.02 | **.460±.001** | 6970±187 | **.429±.001** | **.475±.001** | **.520±.001** | 41.1M |
| COMPLEX-MOS † | 5.93±.17 | .439±.003 | 5310±63.7 | .417±.001 | .446±.004 | .481±.008 | 49.2M |
| RESCAL | 8.14±.96 | .378±.020 | 5750±960 | .357±.018 | .387±.021 | .417±.023 | 63.1M |
| RESCAL-MOS | 6.79±1.58 | .421±.028 | 4820±175 | .394±.032 | .433±.024 | .469±.024 | 71.1M |
| CONVE | **5.55±.01** | .433±.001 | **3860±25.2** | .394±.001 | .448±.001 | .511±.002 | 96.2M |
| CONVE-MOS | 7.26±.15 | .416±.010 | 4400±62.3 | .391±.012 | .427±.009 | .462±.009 | 104.3M |

† Results for COMPLEX use halved embedding sizes $d = 100$ and $d = 500$.

*Table 16.* MRR and NLL on `ogbl-biokg` at $d = 1000$ with different numbers of softmaxes $K$.

| MODEL | $K$ | NLL ↓ | MRR ↑ | MODEL | $K$ | NLL ↓ | MRR ↑ |
|---|---|---|---|---|---|---|---|
| DISTMULT | 1 | 4.89±0.02 | .801±.002 | COMPLEX | 1 | 4.86±0.00 | .806±0.001 |
| DISTMULT-MOS | 1 | 4.42±0.01 | .821±.000 | COMPLEX-MOS | 1 | 4.46±0.01 | .820±0.001 |
| DISTMULT-MOS | 2 | 4.37±0.01 | .831±.001 | COMPLEX-MOS | 2 | 4.41±0.02 | .827±0.002 |
| DISTMULT-MOS | 4 | 4.34±0.01 | .837±.001 | COMPLEX-MOS | 4 | 4.39±0.00 | .836±0.001 |
| DISTMULT-MOS | 8 | **4.33±0.00** | **.841±.001** | COMPLEX-MOS | 8 | **4.37±0.00** | **.838±0.001** |

*Table 17.* Effect of the projection depth and the number of softmaxes $K$ on `ogbl-biokg` at $d = 1000$, for DISTMULT-MOS and COMPLEX-MOS. The *baseline* rows are the bottlenecked models without KGE-MOS. Both a deeper projection (2-layer > 1-layer) and more softmaxes (larger $K$) improve performance.

| | DISTMULT | | | COMPLEX | | | |
|---|---|---|---|---|---|---|---|
| PROJECTION | $K$ | NLL ↓ | MRR ↑ | PROJECTION | $K$ | NLL ↓ | MRR ↑ |
| BASELINE | – | 4.89 | .801 | BASELINE | – | 4.86 | .806 |
| 1-LAYER | 1 | 4.57 | .816 | 1-LAYER | 1 | 4.55 | .815 |
| | 2 | 4.50 | .823 | | 2 | 4.51 | .822 |
| | 4 | 4.45 | .828 | | 4 | 4.50 | .827 |
| | 8 | 4.47 | .833 | | 8 | 4.47 | .832 |
| 2-LAYER | 1 | 4.42 | .821 | 2-LAYER | 1 | 4.46 | .820 |
| | 2 | 4.37 | .831 | | 2 | 4.41 | .827 |
| | 4 | 4.34 | .837 | | 4 | 4.39 | .836 |
| | 8 | **4.33** | **.841** | | 8 | **4.37** | **.838** |

triples that requires an explicit set of negative triples for training, or an estimate thereof. Such negatives are typically unavailable in KGs, which is why SR remains more niche than RR for the datasets we analyse. In contrast, RR and DR only require the observed (positive) triples and are the most common setup for these benchmarks.

Nevertheless, KGE-MOS is readily applicable to SR: the only change is to replace the softmaxes in the mixture with sigmoids, turning the categorical distribution over objects into an independent per-object Bernoulli probability

$$P(y_o = 1 \mid s, r) = \sum_{k=1}^{K} \pi_k(\mathbf{h}_{s,r}) \, \sigma\big(f_k(\mathbf{h}_{s,r})^\top \mathbf{e}_o\big), \quad (14)$$

where $y_o = 1$ if $(s, r, o) \in \mathcal{G}$ and 0 otherwise.

We illustrate this on `ogbl-biokg` with DISTMULT and COMPLEX at $d = 1000$. As negatives are not available in our KGs, we treat all unseen triples as negatives during train-

*Table 18.* Translation- and rotation-based baselines compared with linear-decoder LD-KGEs on `ogbl-biokg` at $d = 1000$. RotatE matches the bottlenecked DISTMULT, and TransE is slightly worse; both are outperformed by DISTMULT-MOS.

| Model | NLL ↓ | MRR ↑ |
|---|---|---|
| DISTMULT | 4.89 | .801 |
| DISTMULT-MOS ($K = 4$) | **4.34** | **.837** |
| TransE | 5.41 | .752 |
| RotatE | 4.57 | .801 |

*Table 19.* Forward-pass slowdown of an amortised ConcatMLP relative to DISTMULT on `FB15k-237` ($d = 200$, batch size $B = 128$), for increasing MLP hidden-layer widths. The overhead is a constant factor – the complexity class is unchanged – but grows with width, while the $(B, |\mathcal{E}|, H)$ activations make memory the binding constraint at scale.

| Hidden layers | Slowdown |
|---|---|
| $[128]$ | $50\times$ |
| $[256]$ | $99\times$ |
| $[512]$ | $188\times$ |
| $[512, 256]$ | $364\times$ |
| $[1024, 512]$ | $931\times$ (OOM on backward) |

ing. This induces a strong class imbalance, which explains the large gap between AUROC and the average precision (AP) / F1 scores in Table 20; both the baseline and the KGE-MOS results could likely be improved with a better negative-sampling strategy. Still, KGE-MOS improves over its bottlenecked baseline on all three metrics, confirming that breaking the rank bottleneck also benefits SR.

*Table 20.* Binary classification (sign reconstruction) results on `ogbl-biokg` at $d = 1000$, obtained by replacing the softmaxes in KGE-MOS with sigmoids. **KGE-MOS improves over its bottlenecked baseline on every metric.**

| Model | AUROC ↑ | AP ↑ | F1 ↑ |
|---|---|---|---|
| DISTMULT | .891 | .065 | .137 |
| DISTMULT-MOS | .985 | .117 | .209 |
| COMPLEX | .932 | .094 | .172 |
| COMPLEX-MOS | **.988** | **.155** | **.244** |

# G. Comparison of KGE-MOS with ensemble models

KGE-MOS is related to ensemble models proposed in, e.g., Wang et al. (2018) for bilinear KGEs. However, the ensemble technique is still bottlenecked as it combines models linearly, and has a high parameter cost as it requires a separate model for each component. In contrast, KGE-MOS uses a single model with several output heads for a low parameter cost, and explicitly uses non-linear transformations to break the rank bottleneck.

Specifically, the ensemble model of Wang et al. (2018) calculates the final scores as

$$\sum_{k=1}^{K} w_k \mathbf{h}_{s,r}^k \mathbf{E}_k^\top$$

with $w_i$ combining a linear renormalisation factor and an ensemble parameter. Compare this with the KGE-MOS scores

$$\sum_{k=1}^{K} \pi_k(\mathbf{h}_{s,r}) \operatorname{softmax}(f_k(\mathbf{h}_{s,r})\mathbf{E}^\top)$$

there are two important differences to highlight:

**On breaking rank bottlenecks:** the ensemble technique is still linear, with a rank bottleneck of $Kd$. Indeed, it can be rewritten as the factorisation

$$[w_0 \mathbf{h}_{s,r}^0; \ldots; w_K \mathbf{h}_{s,r}^K][\mathbf{E}_0^\top; \ldots; \mathbf{E}_K^\top]$$

with $[;]$ a concatenation operation on the first axis. In contrast, KGE-MOS achieves a rank $\gg Kd$ using a fraction of the parameters (see Appendix F.4.2). It uses $K$ softmax layers *within* the mixture, which prevents this factorization and is necessary to break the bottlenecks. Notice also that the mixture weights in KGE-MOS depend on the query embedding, which is not the case for the ensemble model.

**On parameter efficiency:** the ensemble model uses $k = 1, \ldots, K$ output embedding spaces $\mathbf{E}_k$, which costs $K$ times more parameters. In contrast, KGE-MOS uses projections $f_k$ of the query embedding for a low parameter cost. For example, with DISTMULT/COMPLEX, this costs only 4% more parameters in `openbiolink`, the largest dataset used in the experiments.

