# OpenReview forum: "On the Theoretical Limitations of Embedding-based Link Prediction"
_ICML.cc/2026/Conference — ICML 2026 regular_

### Official Review · Reviewer_4oMn · 2026-03-01

**Soundness:** 3
**Presentation:** 3
**Significance:** 2
**Originality:** 3
**Overall Recommendation:** 4
**Confidence:** 4

**Summary:**

This paper studies rank bottlenecks in Knowledge Graph Embedding (KGE) models arising from low-dimensional linear scoring layers, which restrict the expressivity of the output space when the number of entities is large. The authors show that such bottlenecks limit both ranking performance and probabilistic calibration. To address this issue, they propose KGE-MOS, a Mixture-of-Softmaxes output layer that increases representational capacity with modest additional parameters. The paper provides theoretical bounds characterizing these limitations and empirically demonstrates that KGE-MOS improves ranking accuracy and calibration across multiple KGE architectures and datasets.

**Compliance With Llm Reviewing Policy:**

Affirmed.

**Final Justification:**

My final recommendation remains 4 (weak accept). I find the paper technically solid and substantially improved compared with the earlier version I had reviewed. The rebuttal also addressed my follow-up question clearly and helped place the rank bottleneck discussion in relation to prior work on embedding-based retrieval. While the scope is still somewhat restricted to linear-decoder KGEs and some practical trade-offs remain, I believe the contribution is meaningful and likely to be useful to others working in this area.

**Key Questions For Authors:**

Here is one follow-up question I would like to ask:

Is the rank collapse problem of KGE, as identified in this work in essence similar to the previous work regarding "Theoretical Limitations of Embedding-Based Retrieval" [1], in the sense that cosine similarity is also a form of KGE (dot product) with a relation matrix being identical matrix plus additional normalization?



[1] Weller O, Boratko M, Naim I, et al. On the theoretical limitations of embedding-based retrieval[J]. arXiv preprint arXiv:2508.21038, 2025.

**Limitations:**

Yes

**Strengths And Weaknesses:**

I reviewed an earlier NeurIPS submission of this work and raised several concerns at that time, regarding theoretical justification, scope of applicability, and empirical validation. Specifically, my previous concerns are:

1. The scope of this paper is limited to a particular form of KGE methods. The analysis does not apply to translation-based or rotation-based methods such as TransE and RotatE.
2. The design of KGE-MoS lacks theoretical justification.
3. Lack experiments over modern inductive link prediction methods such as CompGCN or NBFNets

In this submission, the author has integrated the changes they made from previous rebuttals: they have strengthened the theoretical motivation for KGE-MoS, provided empirical rank analyses, added CompGCN experiments, and included large-dimension saturation studies. While some limitations remain (e.g., scope restricted to linear-decoder KGEs and practical computational trade-offs), the current version is technically solid and substantially improved.

---

> ### Author Rebuttal · Authors · 2026-03-31
>
> We appreciate that the reviewer recognizes the improvements of our paper, as their previous feedback helped us! We take the reviewing process as an opportunity to improve our work.
>
> If of interest, in our rebuttal with reviewer vMCD we discuss an extension to KGEs that use the euclidean distance (translation- or rotation- based), and in our rebuttal with reviewer LAh1 we conduct more analyses of the computational trade-off.
>
> > Is the rank bottleneck problem similar to what is discussed in “Theoretical Limitations of Embedding-Based Retrieval”?
>
> Absolutely. We briefly refer to this work in the Related Work. Their main theoretical result is about Ranking Reconstruction (RR), though they add a margin constraint where (i) the positive example with the lowest score and (ii) the negative example with the highest score, must differ by at least a certain $\gamma$. In their Appendix D (arxiv version), they consider the less constrained case without $\gamma$ margin, which derives the sign rank result for RR.
>
> In the KGE setting, if the embeddings do not need to be unit vectors (i.e., dot product without normalization), the margin condition and the usual RR condition is the same: one can always scale the factorised solution by an arbitrary scalar until the minimum difference is $\gamma$. However, if the embeddings must be unit vectors (i.e., cosine similarity), then rescaling the sign rank solution is not allowed. In that case, if a margin must be enforced, then one can use the theoretical result of [Weller et al, 2025]. We will explain this in our Related Work.
>
> Finally, it would be interesting to see if a MoS layer empirically helps in large-scale Information Retrieval tasks.

---

> > ### Author Rebuttal · Reviewer_4oMn · 2026-04-01
> >
> > Thank you for the comments and the additional insights on translation-based/rotation-based results. I will maintain my positive recommendation on this work.

---

### Official Review · Reviewer_LAh1 · 2026-03-03

**Soundness:** 3
**Presentation:** 3
**Significance:** 4
**Originality:** 4
**Overall Recommendation:** 5
**Confidence:** 3

**Summary:**

This paper studies the theoretical limits of embedding-based link prediction models that use a linear decoder, as is common in knowledge graph embeddings (KGEs). The authors show that when the embedding dimension is much smaller than the number of entities, the linear output layer creates a rank bottleneck that restricts which rankings, sign patterns, and probability distributions the model can represent. They derive necessary lower bounds on the embedding dimension required to achieve three objectives: ranking reconstruction (RR), sign reconstruction (SR), and distributional reconstruction (DR), under both universality and graph-specific realisability settings. They also provide a sufficient dimension bound for exact RR and SR based on graph connectivity. To address these limitations in practice, they propose KGE-MOS, a mixture-of-softmax output layer that increases expressivity without scaling the embedding dimension. Experiments on several datasets show that KGE-MOS improves ranking and likelihood performance on larger and denser graphs, consistent with the theory.

**Compliance With Llm Reviewing Policy:**

Affirmed.

**Final Justification:**

This paper provides a theoretical analysis of expressivity limits in linear-decoder KGEs and proposes KGE-MOS as a simple and effective fix.

- I find the work technically sound.
- The empirical results support the claims, though the overall impact is somewhat limited by the focus on linear decoders.
- The rebuttal addressed my questions.

Overall, I recommend acceptance, while noting that the gap between theory and practice could be better addressed.

**Key Questions For Authors:**

[Q1] The realisability results rely on sign rank, which is not computable in practice. In your experience, how tight are the connectivity-based upper bounds compared to what is actually needed to fit the training data?

[Q2] KGE-MOS improves performance on large and dense datasets, but it increases training time. Do you have insight into which part of the mixture contributes most to the gains: the additional projections, the mixing weights, or simply the increase in effective rank? An ablation would be helpful.

**Strengths And Weaknesses:**

Strengths:

[S1] The paper gives a clear and general analysis of rank bottlenecks in linear-decoder KGEs. Instead of focusing on one specific model, the results apply to a broad class that includes both bilinear and neural variants. This makes the theory widely relevant.

[S2] The necessary lower bounds for RR, SR, and DR are well motivated and technically solid. The separation between universality and realisability is especially helpful, since it distinguishes unrealistic worst-case guarantees from graph-dependent requirements.

[S3] The connection between graph connectivity and sufficient embedding dimension is interesting. The bound ( d = 2c + 1 ) links structural properties of the dataset to representational limits, which gives useful intuition about why dense graphs are harder.

[S4] The KGE-MOS layer is simple and easy to integrate into existing models. It does not require redesigning the encoder and adds relatively few parameters compared to scaling the embedding dimension.

Weakness:

[W1] Some of the theoretical quantities, such as sign rank, are not computable in practice. While the paper provides upper bounds, these can be very loose. As a result, it is not always clear how actionable the dimension recommendations are for real datasets. Similarly, the sufficient bound based on maximum out-degree can become extremely large for dense graphs. In practice, models are trained with much smaller dimensions and still perform well, so the gap between theory and practice remains significant.

[W2] The experiments mainly evaluate ranking (MRR) and filtered NLL. The SR task is defined formally, but not directly evaluated with a dedicated metric. Is there a reason why the SR task is not evaluated with a dedicated metric?

[W3] The computational cost of KGE-MOS is noticeably higher during training. Although inference time is less affected, the training slowdown may limit applicability in very large-scale settings. What is the computational cost of KGE-MOS during training?

[W4] The paper focuses entirely on linear-decoder models. While this is clearly scoped, it leaves open how the conclusions extend to newer link prediction methods that do not solely rely on a dot-product decoder.

---

> ### Author Rebuttal · Authors · 2026-03-31
>
> We thank the reviewer for their feedback. We appreciate that they valued the wide relevance of our theory, the distinction between universality/realisability and connections graph properties, as well as the simplicity and wide applicability of the KGE-MoS layer.    We hope we addressed the concerns below and are happy to follow up in the discussion phase.
>
> > (W1+Q1) On the computability of sign-rank and its loose upper bound
>
> Indeed, the bounds are loose because they consider a worst-case permutation of the entities. We intended the bounds to help in comparing several datasets' difficulties, but we agree that providing tighter bounds would give a more actionable insight to recommend dimensions.
> **We derived a heuristic to search for a permutation of the entities that tightens the upper bound of Theorem 4.1.** For details on the heuristic, please refer to our first answer to reviewer vMCD who raised a similar concern. After running the heuristic on each dataset, we find the following upper bounds: (counting inverse relations)
>
> | Dataset | Previous bound | Blocks (max) | Blocks (avg) | New bound |
> | --- | --- | --- | --- | --- |
> | FB15K-237 | 8729 | 592 | 3.3 | 1185 |
> | Hetionet | 30073 | 3552 | 21.7 | 7105 |
> | ogbl-biokg | 58657 | 3025 | 29.3 | 6051 |
> | openbiolink | 36841 | 4235 | 13.7 | 8471 |
>
> **The new bounds are \~5x tighter than our previous ones. We added them in the paper**, with a detailed description of the heuristic in a new appendix. The new bounds are closer to values realistically used in practice, as suggested by the reviewer. Future work could further tighten the bounds by identifying better heuristics.
>
> > (W2) Sign Reconstruction (SR) task evaluation
>
> We have run new binary classification results on `ogbl-biokg` with DistMult and ComplEx at d=1000. To use KGE-MoS in a binary classification setting, the only requirement is to *replace softmaxes with sigmoids in the mixture*.
>
> |  | AUROC | AP | F1 |
> |---|---|---|---|
> | DistMult | .891 | .065 | .137 |
> | DistMult-MoS | .985 | .117 | .209 |
> | ComplEx | .932 | .094 | .172 |
> | ComplEx-MoS | .988 | .155 | .244 |
>
> We initially focused on evaluating RR/DR because this is the most common setup for the datasets we analyzed. SR typically needs an explicit set of negatives for training, or an estimate thereof, which is why it remains more niche compared to RR in KGs. As negatives are not available in our KGs, we have treated all unseen examples as negatives. This leads to a strong class imbalance during training, explaining the large gap between AUROC and AP/F1. One can likely improve both baseline and MoS results with a better negative sampling strategy.
>
> Still, we agree the SR results are interesting to report and show that **KGE-MOS is easily applied to SR**. We thank the reviewer for this suggestion. **We will add the results in an Appendix**.
>
> > (W3) Total training time comparison
>
> On our largest dataset (`openbiolink`), the MoS variants are ~2x slower to complete 30 epochs of training. We report here the baseline with the biggest difference (DistMult) and the one with the smallest difference (ConvE):
>
> * DistMult: 21h22m
> * DistMult-MoS: 43h03m
> * ConvE: 25h25m
> * ConvE-MoS: 47h11m
>
> **This matches the trend of our minibatch measurements (Appendix 7.4.3, Table 8\)** where we reported 1.69-2.75x slower passes per minibatch. **We will add the total training time in Table 8 as an additional insight.**
>
> > (Q2) More ablation on KGE-MoS
>
> We have extended our Table 4 (ablation of number of softmaxes $K$ on the two best performing models for ogbl-biokg, $d=1000$) to also compare MoS with a 1-layer projection instead of the 2-layer projection we used (see Section 5).
>
> The results show that both (i) **improving the projection function helps** (MoS 2-layer \> MoS 1-layer), (ii) **increasing the number of softmaxes helps** (higher K \> lower K). While the mixture is important to enable the geometric non-linearity, it’s also important to have good projections for each mixture head, enabling more precision in the mixed distributions.
>
> | Baseline | Projection | K | NLL ↓ | MRR ↑ |
> |---|---|---|---|---|
> | DistMult | - | - | 4.89 | .801 |
> | DistMult-MoS | 1-layer | 1 | 4.57 | .816 |
> |  |  | 2 | 4.50 | .823 |
> |  |  | 4 | 4.45 | .828 |
> |  |  | 8 | 4.47 | .833 |
> | DistMult-MoS | 2-layer | 1 | 4.42 | .821 |
> |  |  | 2 | 4.37 | .831 |
> |  |  | 4 | 4.34 | .837 |
> |  |  | 8 | 4.33 | .841 |
> | ComplEx | - | - | 4.86 | .806 |
> | ComplEx-MoS | 1-layer | 1 | 4.55 | .815 |
> |  |  | 2 | 4.51 | .822 |
> |  |  | 4 | 4.50 | .827 |
> |  |  | 8 | 4.47 | .832 |
> | ComplEx-MoS | 2-layer | 1 | 4.46 | .820 |
> |  |  | 2 | 4.41 | .827 |
> |  |  | 4 | 4.39 | .836 |
> |  |  | 8 | 4.37 | .838 |
>
> > (W4) Discussions of other link prediction methods not relying on dot-product decoders
>
> If of interest, please refer to our brief discussions with Reviewer vMCD for translation- or rotation-based models, and with Reviewer 2hKw for non-linear models.

---

> > ### Author Rebuttal · Reviewer_LAh1 · 2026-04-01
> >
> > Thank the authors for the rebuttal. My questions have been addressed. I have updated my score.

---

### Official Review · Reviewer_2hKw · 2026-03-09

**Soundness:** 3
**Presentation:** 3
**Significance:** 3
**Originality:** 3
**Overall Recommendation:** 5
**Confidence:** 2

**Summary:**

This paper studies the rank bottleneck in knowledge graph embedding (KGE) models with linear output layers. The authors first derive lower bounds on the embedding dimension required under different reconstruction targets. They then provide sufficient dimension conditions for sign and ranking realisability. Finally, they propose KGE-MOS, a mixture-of-softmaxes output layer designed to alleviate the rank bottleneck, and show empirical improvements on larger benchmark datasets.

**Compliance With Llm Reviewing Policy:**

Affirmed.

**Final Justification:**

The rebuttal has fully addressed my concerns, so I maintain the positive score.

**Key Questions For Authors:**

1. The current analysis focuses on linear-output KGEs. How would the main conclusions change if one used a more expressive nonlinear scorer, such as an MLP over concatenated source and target representations? Would such a parameterization already avoid the rank bottleneck identified in this paper? If so, what would be the main advantage of KGE-MOS compared with these alternatives?

**Limitations:**

yes

**Strengths And Weaknesses:**

### Strengths

1. **Clear problem formulation.**
   The paper provides a clear and well-structured formulation of the rank bottleneck problem in linear-decoder KGEs. In particular, the distinctions among the different reconstruction objectives are presented clearly, and the corresponding dimensionality requirements are analyzed in a systematic way.

2. **Strong theoretical contribution.**
   The theoretical analysis is one of the main strengths of the paper. The authors derive both lower and upper bounds on the required embedding dimension, and they also discuss the practical implications of these bounds in realistic graph settings.

3. **Empirical validation.**
   The proposed method shows consistent improvements on real-world benchmark datasets, especially on larger graphs where the bottleneck is expected to be more severe. This provides supporting evidence for the practical relevance of the theoretical analysis.

### Weaknesses

1. **The scope of the theoretical analysis is somewhat narrow.**
   The paper focuses primarily on KGE models with linear output layers. While this setting is well-motivated, it is not entirely clear how restrictive this assumption is in practice. In particular, one natural question is whether a simple modification of the output parameterization could already provide substantially stronger expressivity. For example, if one scores each source-target pair using a small MLP over their concatenated representations, would it be possible to achieve the same reconstruction goals with much lower embedding dimension? If so, it would be helpful for the paper to more clearly articulate the advantages of KGE-MOS relative to such more general nonlinear scoring approaches, in terms of either expressivity, efficiency, or compatibility with the original KGE framework.

2. **Presentation issues.**
   Some subsection titles in Section 3 begin with ellipses (“...”), which appears to be a formatting issue and should be corrected.

---

> ### Author Rebuttal · Authors · 2026-03-31
>
> We are happy that the reviewer appreciates the clarity of our paper, the usefulness of our systematic theoretical formulation, and the empirical validation! We will fix the formatting in the camera-ready.
>
> > comparison with nonlinear decoders like MLPs over the concatenated representations of s,r,o
>
> Yes, an MLP over concatenated representations would avoid rank bottlenecks, as it can represent arbitrary score functions. The problem is that they do not scale well for object prediction $(s,r,?)$. From the Related Work section, when talking about one such model (NBFNet \[1\]): “*Unlike the models studied in our paper, NBFNet non-linearly encodes subject and object jointly and works well in recent tests. However, it is challenging to scale to large datasets.*”
>
> To illustrate this, we implemented a ConcatMLP baseline which concatenates $s,r,o$ embeddings and uses a two-layer MLP to project the representation to a score, as suggested by the reviewer. While LD-KGEs only need one pass over a linear projection to score all objects, ConcatMLP needs to do $|\\mathcal{E}|$ passes over the MLP network. It is possible to batch some of the objects together to parallelize the call, but the batch size $b \\ll |\\mathcal{E}|$ is limited by GPU memory. Typically, many neural network passes are still needed per query.
>
> We measured how long it takes to score objects given a minibatch of subject and relation queries on FB15K-237. **On our smallest dataset, we find that ConcatMLP is already \~10000 times slower** than DistMult. This is averaged over 20 runs at d=200. **At d=1000, the backward pass of ConcatMLP could not be computed due to memory constraints**; one would need to sample only a few objects per training pass.
>
> | Model | Minibatch Fwd | Minibatch Fwd + Bwd |
> |--------|----------|---------|
> | DistMult   | 0.230ms      | 1.180ms    |
> | ConcatMLP| 2595.12ms    | 10614.5ms   |
>
> This is why **such nonlinear models are typically used for classifying only a few triples or for relation prediction tasks** (smaller output space). **For object prediction, an approach like MoS is preferable for the large KGs we study**. MoS is a drop-in replacement that preserves efficient vector-matrix multiplications for each component.
>
> \[1\] *Neural Bellman-Ford Networks, Z Zhu et al, 2022*

---

> > ### Author Rebuttal · Reviewer_2hKw · 2026-04-01
> >
> > Thanks for the rebuttal, which has addressed my concerns. Therefore, I maintain my positive score.

---

### Official Review · Reviewer_vMCD · 2026-03-13

**Soundness:** 3
**Presentation:** 3
**Significance:** 3
**Originality:** 3
**Overall Recommendation:** 5
**Confidence:** 3

**Summary:**

The paper studies the theoretical expressiveness limits of knowledge graph embedding models with linear decoders, arguing that the main bottleneck lies in the output layer rather than only in the encoder. It derives lower bounds on the embedding dimension required for ranking, sign, and distributional reconstruction, and relates these limits to graph structure through realizability and connectivity-based analysis. To address this bottleneck, the paper proposes KGE-MOS, a mixture-of-softmax decoder inspired by language models.

**Compliance With Llm Reviewing Policy:**

Affirmed.

**Final Justification:**

Please see strengths and rebuttal acknowledgement comments.

**Key Questions For Authors:**

As also mentioned in the weaknesses the realisability results depend on sign rank which is NP-hard, does such bound have any practical value essentially? are there any better proxies for such a result that could be practically computed?

Can the authors better clarify whether the sufficient bound is intended only as a qualitative explanation of graph difficulty?

**Limitations:**

yes

**Strengths And Weaknesses:**

* Strengths

The paper provides a theoretical analysis on the expressiveness of linear-decoder KGEs and argues that the bottleneck actually exists in such linear decoders rather than weak encoders. This is a promising perspective especially in the era of massive models that can direct the analysis to more expressive decoders rather than to bigger models. KGEs are a particularly relevant setting because of the large entity spaces and complex relations they must represent.

There is a lot of rigour in the analysis opossite to most studies that collapse everything into ranking metrics the authors include both sign reconstruction to distinguish possible versus non-possible triples as well as distributional reconstruction. Table 1 makes the argument quite clear regarding the necessary embedding dimensions for expressing each task objective for LD-KGEs.

The experimental set-up and results mirrors exactly the theoretical claims made and shown by the authors. They successfully show the regimes where the theory works while the sufficient condition based on maximum out-degree is practically useful and elegant.

* Weaknesses

The realisability bounds rely on sign rank, which is NP-hard to compute. This substantially limits their practical usefulness, since the theory does not provide a usable procedure for selecting the embedding dimension on a real dataset.

The sufficient bound summarized in Table 2 appears too loose to be operationally meaningful, as the resulting dimensionalities are far beyond what would be realistic in practice. As a consequence, the bound is more informative as an explanation of dataset structure than as a concrete prescription for model design.

The empirical comparison also remains somewhat narrow. Although the paper is explicitly focused on linear-decoder KGEs, including representative translation- or rotation-based baselines, such as TransE [1], would have helped clarify whether the proposed decoder-side remedy remains competitive beyond the specific model class covered by the theory.

[1] Translating Embeddings for Modeling Multi-relational Data, Bordes et al.

---

> ### Author Rebuttal · Authors · 2026-03-31
>
> We thank the reviewer for their feedback, and appreciate that they valued the relevance of our new perspective for KGEs, the rigor of our analysis and the empirical validation.
> We hope we addressed the remaining concerns below and are happy to follow up in the discussion phase.
>
> > On the computability of sign-rank and its loose upper bound
>
> Indeed, computing the sign rank is intractable, and estimating it efficiently is an open problem. Recent approximations \[1\] do not scale to large KGs. This is why the sign-rank upper bound from Section 4 is important. However, we agree with the reviewer (and reviewer LAh1), that the provided bound is loose and only gives a rough insight comparing datasets' difficulties rather than suggesting actual dimension choices. This is because it considers a worst-case ordering of the entities.
>
> To improve the upper bound, **we can heuristically search for a permutation of the entities that minimizes the sign changes in the rows of** $Y^\\pm$. Given a good permutation of entities, the \+1 entries in the proof of the upper bound (Appendix B, Figure 4\) should form *contiguous blocks*. The dimensional requirement becomes the maximum number of disjoint blocks across (subject,relation) rows. We derived a simple approach to permute entities such that entities that co-occur often as objects of the same subject-relation pairs, are contiguous in the permutation. We build a matrix of objects vs objects, where entries count in how many $(s,r)$ pairs they co-occur. Then, we run a breadth-first search (Reverse Cuthill-McKee heuristic) to find a permutation minimizing sparse blocks.
>
> Concretely, after running the heuristic on each dataset, we find the following upper bounds: (counting inverse relations)
>
> | Dataset | Previous bound | Blocks (max) | Blocks (avg) | New bound |
> | --- | --- | --- | --- | --- |
> | FB15K-237 | 8729 | 592 | 3.3 | 1185 |
> | Hetionet | 30073 | 3552 | 21.7 | 7105 |
> | ogbl-biokg | 58657 | 3025 | 29.3 | 6051 |
> | openbiolink | 36841 | 4235 | 13.7 | 8471 |
>
> **The new bounds are \~5x tighter than our previous ones.** **We added the above table to the paper**, with a detailed description of the heuristic in the appendix. These dimensions come from interpolated solutions and may be hard to learn via gradient descent (see discussion at the end of Section 4). One may need to overshoot the dimension for models learned by gradient descent. Still, **this gives a better quantitative insight, as suggested by the reviewers, closer to dimensions used in practice.**
>
> \[1\] *Sign rank versus VC dimension, N Alon et al, 2015*
>
> > Include a representative example of translation- or rotation- based baselines, such as TransE
>
> We extended our discussion of translation- or rotation-based baselines with the following, and measured TransE and RotatE baselines on `ogbl-biokg` (they will be added in an Appendix).
>
> While structured bounds for universality/realizability in RR/DR/SR remain to be explored, we know that the lower bound for universality in SR will be at least $\\mathcal{E}-1$. This comes from the VC dimension of Euclidean distance models \[2\], which tells us that to shatter (correctly classify) an arbitrary combination of $|\\mathcal{E}|$ objects, such models need $d=\\mathcal{E}-1$. This gives only a sanity check for universal SR and is likely not enough: universality requires a model capable of shattering *all* arbitrary combinations of objects at the same time, not just one. Still, the sanity check dimension is already close to the one for linear decoder KGEs. In other words, **on the one known result (SR universality), changing the geometry of the score function to an Euclidean one does not improve expressivity.**
>
> Empirically, as expected, we find that **RotatE performs comparatively to the bottlenecked linear-decoder KGEs on ogbl-biokg** (TransE is slightly worse).
>
> | ogbl-biokg (d=1000) | NLL ↓ | MRR ↑ |
> | --- | --- | --- |
> | Distmult | 4.89 | .801 |
> | Distmult-MoS (K=4) | 4.34 | .837 |
> | TransE | 5.41 | .752 |
> | RotatE | 4.57  | .801 |
>
> Along with new theoretical results, it would be interesting for future work to investigate a translational extension of MoS, e.g.
> $$\\sum\_{k=1}^{K} \\pi\_k(h\_{s,r})\\, \\operatorname{softmax}(- \\|f\_{k}(h\_{s,r}) \- E^\\top\\|^2\_d) \+ \\tau\_{s,r}$$
> Where $f\_{k}(h\_{s,r}) \- E^\\top$ is “broadcasted” and $\\|\\cdot\\|^2\_d$ is the L2 norm taken over the feature axis.
>
> \[2\] *Balls in R^k do not cut all subsets of k \+ 2 points, RM Dudley, 1979*. This paper is about hypersphere classifiers, where classes are defined by a center and a radius. In models like TransE, the embedding of each entity acts as a center, and the radius is some threshold.

---

> > ### Author Rebuttal · Reviewer_vMCD · 2026-04-02
> >
> > I would like to thank the authors for their response. All my concerns have been addressed and thus I move towards a clear acceptance of the paper.

---

### Decision · Program_Chairs · 2026-04-30

**Decision:**

Accept (regular)

**Comment:**

Reviewers agree that the paper makes a good technical contribution analyzing the theoretical limits of linear-decoder KGEs, and appreciate the connections made with graph size and connectivity. The proposed non-linear output models (KGE-MOS) improves ranking accuracy in experiments.

Some concerns were raised, especially around the limited practical implications of the analysis (due to the use of quantities such as sign-rank that are not computable in practice, or excessively loose bounds), the somewhat limited scope (especially the linear output assumption), and the narrow empirical evaluation. Most of the technical concerns were addressed during discussion, for example by providing improved bounds based on heuristics, and additional experiments. The authors are encouraged to add as much detail as possible to the revision.

In particular, one argument made during the rebuttal (to justify the lack of nonlinear baselines) that ConcatMLP is ~10000 times slower than DistMult on one of the datasets, may be using an overly naive implementation. Encoder outputs can be cached and should not be systematically recomputed, only the MLP part needs to be. A more careful argument/estimate needs to be made here, and we trust that the authors will give a more nuanced analysis in the revision. They are strongly encouraged to include nonlinear baselines in the experiments.